# A QUANTITATIVE ANALYSIS OF SEMANTIC INFORMATION IN DEEP REPRESENTATIONS OF TEXT AND IMAGES

## ABSTRACT

Deep neural networks are known to develop similar representations for semantically related data, even when they belong to different domains, such as an image and its description, or the same text in different languages. We present a method for quantitatively investigating this phenomenon by measuring the relative information content of the representations of semantically related data and probing how it is encoded into multiple tokens of large language models (LLMs) and vision transformers. Looking first at how LLMs process pairs of translated sentences, we identify inner "semantic" layers containing the most language-transferable information. We also identify layers encoding semantic information within visual transformers. We show that caption representations in the semantic layers of LLMs predict visual representations of the corresponding images. We observe significant and model-dependent information asymmetries between image and text representations.

## 1 INTRODUCTION

Current research explores the conjecture that Large Language Models (LLMs) possess a *universal language of thought*, enabling them to process concepts across different languages (Peng and Søgaard, 2024; Lindsey et al., 2025; Huh et al., 2024; Singh et al., 2019; Brinkmann et al., 2025). A compelling example comes from Anthropic's study of the Claude model (Lindsey et al., 2025), which shows that the network occasionally reasons within a shared conceptual space across languages. Huh et al. (2024) shows that representations of many different LLM/ViTs processing semantically equivalent data are similar to each other, and that this form of alignment is correlated with model size. More generally, representations in deep networks, including but not limited to LLMs, are converging, potentially pointing toward shared underlying conceptual structures.

In this work, *we seek to turn the qualitative notion that deep networks develop semantically meaningful shared representations into quantitative measures, identifying where in the network, and to what extent, semantic information is encoded*. We can cast the identification of shared semantic content as a problem of relative information: different networks learn distributions defined on different data domains, such as images and captions. Those distributions loosely span a common underlying subspace, associated with shared semantic information. The support of the different distributions does not reduce to this subspace, but spans other subspaces, which are data and architecture specific. To quantify relative information we need a similarity criterion that is both (i) asymmetric, as there may be a partial order relation between models and representations (qualitatively, the subspaces which are data specific might have different dimensions); (ii) computationally efficient for representations of dimension of order $10^5 - 10^6$, which is the typical number of neurons in a deep representation. Cross-entropy measures how difficult it is to encode an event from a distribution into another and would be the ideal choice, but estimating it is computationally difficult due to the dimension of the representations. On the other hand, local methods like the average number of nearest neighbors (Doimo et al., 2020; Huh et al., 2024) fail to capture the asymmetry between spaces. To overcome the limitations, we employ the Information Imbalance (Glielmo et al., 2022): a method that leverages conditional ranking to provide an asymmetric measure of relative mutual predictivity, which has been shown to be an excellent proxy of the cross entropy (Del Tatto et al., 2024).

[1] These authors contributed equally to this work.

We analyze the representation of translations of the same sentences in different languages, using one of the most powerful open-source language models, DeepSeek-V3 with 671B parameters (DeepSeek-AI et al., 2025), comparing it to the mid-sized Llama3.1-8b LLM (Meta, 2024). Moreover, we analyze the deep representations of pairs of images and of human-generated descriptions, and of pairs of images depicting the same object, with the goal of capturing the alignment between image and caption representations in different architectures and the relative predictive power of images and texts. As visual models, we employ image-gpt-large (Chen et al., 2020) and DinoV2-large (Oquab et al., 2024).

Our main contributions are as follows:

- We analyze the DeepSeek-V3 language model and measure, layer by layer, the relative information content between representations of sentence pairs that are translations of each other. This reveals a broad region of the network, robust across language pairs, that consistently encodes shared semantic content.

- We compare different summaries of the data -considering concatenations, mean and last tokens- to quantify their impact on similarity scores. We find that considering several tokens yields significantly stronger similarity, suggesting that semantic information is spread across many tokens.

- We also identify semantic layers in vision transformers by processing pairs of images that share the same class. These semantic layers are also the most informative about DeepSeek-V3 representations of descriptive image captions, where we observe significant information asymmetries between image and text representations.

**Related work.** The presence of semantic layers containing shared information in deep networks is connected to the notion of representation alignment (Sucholutsky et al., 2024). Two different models can be 'stitched' together with (Bansal et al., 2021) or without (Moschella et al., 2023) a trainable stitching layer at several different stages of the network. For instance, two vision models can effectively use similar low-level features (Lenc and Vedaldi, 2019). Nonetheless, the types of features that we probe for in a semantic layer are instead abstract, encoding the collective relationships between the input constituents, independently of the particular way in which they were expressed in the specific input modality at disposal. The general representations we are probing lie behind the so-called *Platonic representation hypothesis*, postulating that, as model quality improves, models converge to similar representations (Huh et al., 2024). We significantly extend our understanding of the nature of these shared representations with respect to (Huh et al., 2024), since for autoregressive language models, the authors take the average across the token axis, reducing the dimensionality of the problem, whereas for vision models, they only consider the "class" token. We address instead where semantic information is encoded within the network, namely across different tokens and different layers. LLM representations have also been studied in Cheng et al. (2025) using, among other techniques, the Information Imbalance. This work is focused on the analysis of the last-token representation of the same text in different models, and not on multimodal data.

Other earlier works point to deep similarities across architectures and modalities (Kornblith et al., 2019; Sorscher et al., 2022; Maniparambil et al., 2024). Again, these works do not focus on determining the location and nature of the shared semantic layers, nor do they include a latest-generation, large model such as DeepSeek-V3. Our work also relates to studies that probe the extent to which multilingual LLMs produce shared representations across languages (and whether this is due to English acting as a pivot language), e.g. (Peng and Søgaard, 2024; Singh et al., 2019; Brinkmann et al., 2025; Wendler et al., 2024; Zhao et al., 2024). In particular, Wendler et al. (2024) analyzes whether multilingual models align non-English inputs by implicitly routing them through English, focusing on logit lens experiments. To this research line, we contribute a method to compute similarity that allows us analyze long spans of tokens and to capture asymmetry between language-specific representations, without the use of the embedding or unembedding matrices. Finally, Chughtai et al. (2023) also study universality in learned representations, but while their analysis is purely theoretical and operates in controlled toy settings, our work examines analogous questions using real-world multilingual and multimodal data on actual state-of-the-art LLMs and ViTs.

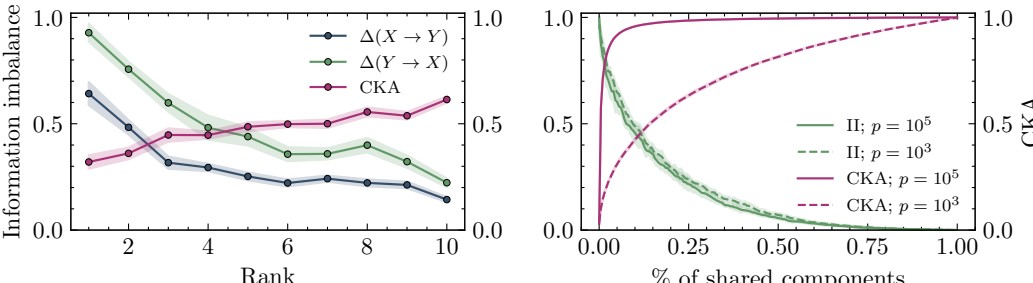

Figure 1: **Left:** Information Imbalance $\Delta(X \to Y)$, $\Delta(Y \to X)$, compared with Central Kernel Alignment (CKA), for a synthetic Gaussian construction in which each index $r$ generates a pair $(X_r, Y_r)$ via $Y_r = B_r X_r + \varepsilon$, with $X_r \sim \mathcal{N}(0, I)$ and $\varepsilon \sim \mathcal{N}(0, \sigma^2 I)$, in $D = 10$ dimensions. The matrices $B_r \in \mathbb{R}^{D \times D}$ are designed to have monotonically increasing rank—from rank one at $r=1$ to full rank at the final index. **Right:** Statistical-power benchmark on a high-dimensional Gaussian model. We compute the Information Imbalance and CKA using only a fraction of the $p$ components, ranging from small subsets to the full vector for $p = 10^3$ and $p = 10^5$. In both figures, we report the standard error computed by averaging over ten jackknife repetitions.

## 2 THE INFORMATION IMBALANCE

Large language models encode information in high-dimensional spaces, transforming representations across layers to perform a task. The Platonic representation hypothesis (Huh et al., 2024) suggests that –for relevant model sizes– representations converge to similar neighboring structures regardless of (i) the task of the model and (ii) the specific encoding of the information. The emerging regularity of these structures underpins the idea that meaning drives their organization, with models loosely acting as continuous mappings from a hidden manifold in which semantically similar ideas are close to each other. We can thus cast the convergence to a universal representation as a mutual information problem. If we model pairs of representations as random variables with a joint probability distribution, for instance an image and its caption, we can measure the information lost by replacing the joint distribution $p(x, y)$ with the product of the marginals $p(x)p(y)$. If the distributions are independent, observing captions is not informative of the density of images. If representations converge, knowing a caption is highly informative and the joint distribution is not approximated well by the marginals.

Mutual information alone, however, is not sufficient to compare different models. The Platonic hypothesis argues that convergence towards a universal set of representations depends on model size. We therefore need to determine (i) whether the effect correlates with the specific encoding of the information (images, captions, . . .); (ii) whether architectural differences matter; and (iii) how convergence changes with model size. From an information-theoretic standpoint, we need an asymmetric measure that quantifies how much uncertainty over the representation in $Y$ (e.g., captions) remains once we observe the representation in $X$ (e.g., images). In practice, estimating information-theoretical measures which could quantify such asymmetry is difficult in the high-dimensional spaces produced by modern models. For this reason, we turn to the Information Imbalance (Glielmo et al., 2022; Del Tatto et al., 2024), a rank-based measure that tests whether neighbourhoods in one representation space predict neighbourhoods in another: if representations converge, the nearest neighbours of a point in $Y$ should also be nearest neighbours of the corresponding point in $X$ in the other architecture.

Formally, the Information Imbalance compares the neighborhood structures of two representation spaces $X$ and $Y$. Given representations $\{\mathbf{z}_i^X\}_{i=1}^N$ and $\{\mathbf{z}_i^Y\}_{i=1}^N$, we compute all pairwise distances and rank points $j \neq i$ in each space by increasing distance, obtaining ranks $r_{i,j}^X$ and $r_{i,j}^Y$. The Information Imbalance from $X$ to $Y$ is the normalized average rank in $Y$ of the nearest neighbor of $i$ in $X$:

$$\Delta(X \to Y) = \frac{2}{N - 1} \frac{1}{N} \sum_{i=1}^{N} r_{i,\text{NN}(i)}^Y,$$

where $\mathrm{NN}(i)$ denotes the index of the closest point to $i$ in space $X$, i.e. $r^X_{i,\mathrm{NN}(i)} = 1$. If neighborhoods in $X$ predict neighborhoods in $Y$, these ranks are small and $\Delta(X \to Y)$ is close to zero; if $X$ carries no information about $Y$, the expected rank is uniform and $\Delta(X \to Y) \approx 1$. The asymmetry $\Delta(X \to Y) \neq \Delta(Y \to X)$ quantifies directional predictability across models or modalities. In Fig. 1 (left), we show that the information imbalance captures both the strength and the direction of the relative information across layers in a controlled synthetic setting, highlighting how asymmetry emerges when information is compressed or degraded. We compute the Information Imbalance $\Delta(X \to Y)$ and $\Delta(Y \to X)$ for a synthetic Gaussian construction in which each index $r$ generates a pair $(X_r, Y_r)$ via $Y_r = B_r X_r + \varepsilon$, with $X_r \sim \mathcal{N}(0, I)$ and $\varepsilon \sim \mathcal{N}(0, \sigma^2 I)$, in $D = 10$ dimensions. The $B_r$s are defined by first choosing a target rank $r \in \{1, \ldots, D\}$ and then sampling independent Gaussian factors $U_r \sim \mathcal{N}(0,1)^{D \times r}$ and $V_r \sim \mathcal{N}(0,1)^{r \times D}$, with the linear map given by $B_r = U_r V_r$, which enforces a transformation of rank $r$. High–rank settings preserve most of the structure in $X_r$, whereas low–rank settings collapse information into a low-dimensional, noisy subspace. The information-imbalance curves reveal the resulting *directional* improved predictivity, showing that $\Delta(X \to Y)$ is lower than $\Delta(Y \to X)$. For comparison, we also report CKA, which tracks the overall loss of alignment between $X_r$ and $Y_r$ but, being symmetric, cannot expose the asymmetry captured by Information Imbalance. A complementary statistical-power benchmark is shown in Fig. 1 (right), where we compute the Information Imbalance between a gaussian vectors with $p$ components and a vector containing only given a fraction of them, for cases $p = 10^3$ and $p = 10^5$. For a $10^5$-dimensional Gaussian, CKA saturates to 1 even when only a small fraction of the components is observed, while the Information Imbalance retains discriminative resolution up to a much larger fraction of shared features. This illustrative example echoes the findings of Huh et al. (2024), that showed their neighborhood overlap was preferable to CKA in high dimensions.

Our representations live in extremely high-dimensional spaces, with embedding dimensions of a few thousand across 40–1024 tokens, leading to feature spaces of size up to $\mathcal{O}(10^6)$. A central difficulty is that it is not clear a priori which summary of these token-level representations should be used: many studies rely on the last token or on the average, but these choices may discard information. One advantage of the Information Imbalance is that it scales to high dimensions, allowing us to concatenate all tokens and systematically compare different summaries without committing to a specific one. To reduce memory usage and stabilise distance computations, we binarize activations using the sign function—a quantization method widely used in neural network training (Hubara et al., 2018; Guo, 2018; Hubara et al., 2016; Wang et al., 2023) and in representation analysis, where it approximately preserves angles (Anderson and Berg, 2018) and has proved suitable for studying the intrinsic dimension of many-token representations (Acevedo et al., 2025). Standard practice in representation analysis involves clipping activations and normalising vectors to mitigate outlier dimensions (Kovaleva et al., 2021; Bondarenko et al., 2023; Sun et al., 2024); binarization implicitly performs both steps by clipping values to $\pm 1$ and enforcing equal norms. In Sec. G of the Supp. Inf., we show that binarized and floating-point activations yield essentially identical information-imbalance curves, indicating that our results depend on the geometric arrangement of points—relative distances and neighbourhoods—rather than on the precise numerical values of the activations.

## 3 RESULTS

### 3.1 THE LOCAL GEOMETRY OF LLM REPRESENTATIONS OF TRANSLATED SENTENCES ALLOWS THE IDENTIFICATION OF SEMANTIC LAYERS

We consider sets of features $A_l$ and $B_l$ in two different languages. The index $l$ labels different layers and ranges from 1 to $L$. If a sentence in the language associated with space $A_l$ is tokenized into $T$ tokens, then at each layer $l$, $A_l \in \mathbb{R}^{T \times E}$, where $E$ denotes the embedding dimension of the language model processing the sentence. What we refer to as the semantic content (or meaning) of a sentence, is a global property, emerging from the interaction of all the symbols present in it. However, in the deep representations the attention mechanism can move information across the tokens, and eventually concentrate it in a specific token, as indeed happens by construction in the last layer of causal models. One of the goals of this work is studying explicitly where the information is stored in the deep layers. This will be addressed in Section 3.1.2. We use human-made translations from Helsinki-NLP/opus_books. Data processing details are in Supp. Inf. C.

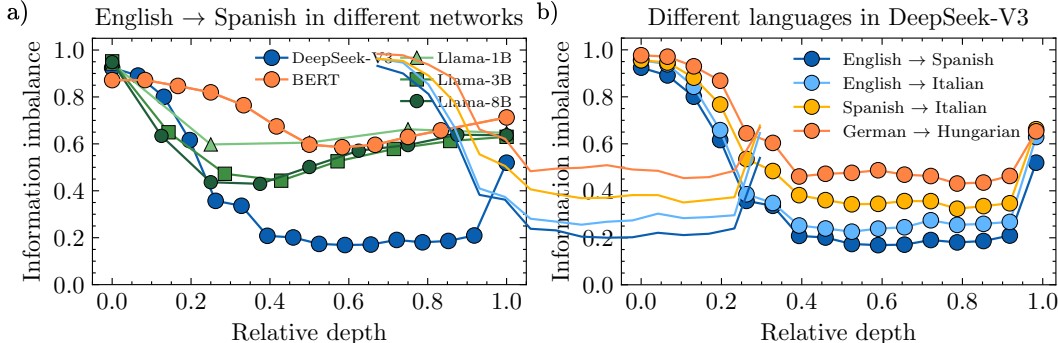

Figure 2: **Panel a)** Information Imbalance from English to Spanish, using representations generated at equal depth from translated sentences of opus books, as a function of the relative network depth, for DeepSeek-V3, BERT-multilingual and Llama3 with 1, 3, and 8 billion parameters. A smaller value of the Information Imbalance correspond to higher predictive power. We used the concatenation of the last 20 tokens for the computation. **Panel b)** Information Imbalance between equal-layer representations of DeepSeek-V3 processing different translation pairs. For each language pair in the legend, we show the Information Imbalance from the first language to the second. The Jackknife error bars (five repetitions subsampling 2,000 of 5,000 sentences) are too small to show; the marker sizes act as an upper bound on these errors.

Starting with English-Spanish translations, we compute the Information Imbalance between $A_l$ and $B_l$ for every layer in the network, excluding the embedding layer, using the last 20 tokens of each representation (in Sec. 3.1.2 we study the dependence of the Information Imbalance on the number of tokens). Note that $A_l$ and $B_l$ are generated independently, with the network reading one language at a time, without any specific prompt or instruction. We also verified that our conclusions are robust to standard preprocessing and similarity metrics, such as clipping-and-normalization and the neighborhood overlap of Huh et al. (2024), as detailed in Supp. Inf. F.

Results are shown in panel a) of Fig. 2. Since the initial layers of LLMs must process the input, we expect the corresponding representations to be strongly language-dependent, and hence not highly informative about each other. As we consider deeper layers, we observe that they become progressively more informative about the deep representations in the other language. Since the information shared by both input languages must be semantic in nature, this analysis confirms that the data structure of deep representations seems to encode information in a universal, language-independent manner. In the very last layers of DeepSeek-V3, we see an apparent jump in Information Imbalance. Part of this effect is due to the fact that we report only a subsample of layers for computational reasons, which makes the transition appear more abrupt. Beyond this sampling artefact, the increase is expected: the final layers are driven by next-token prediction and therefore encode strongly language-specific representations, which cannot be universal.

Note that -under the Platonic representation hypothesis- for an ideal dataset of translations, an ideal network separately processing the sentence pairs should generate equivalent representations, namely somewhere in the network the Information Imbalance from one language to the other should be close to zero. Thus, we define semantic quality in terms of mutual predictability, which should manifest as a lower Information Imbalance. In Fig. 2, for English-Spanish pairs the Information Imbalance of DeepSeek-V3's representations has a very broad minimum of order 0.2, roughly between relative depths 0.4 and 0.9. This number is impressively small, considering that the dimension of the representations which are compared is $20 \cdot 7000 \sim 150,000$. As a comparison, if the data were generated by Gaussian processes, one would observe an Information Imbalance of 0.2 if approximately 20% of the features were shared between the two representations (see Fig. 1). Instead, for Llama3.1-8b the minimum Information Imbalance is significantly higher, of order 0.3, corresponding to approximately 15% shared features. Moreover, the minimum is narrower, around a relative depth of 0.3, suggesting that bigger and better models generate representations of higher semantic content.

Finally, to account for different model sizes, we performed the same experiments on two additional Llama models, observing that the minimum Information Imbalance moves downward and gets slightly displaced toward earlier layers. Notably, this effect only takes place in inner layers, whereas the II measured in both initial and final layers does not seem to scale down with model size. To account for a difference in training objective, we also added a multilingual BERT model with 0.2B parameters, also showing a qualitatively similar behavior.

We also measure the intrinsic dimensionality of DeepSeek-V3's representations using the BID estimator (Acevedo et al., 2025). We find two local maxima of roughly the same height that coincide with the beginning (around 0.4) and the end (around 0.9) of the semantic region observed in Fig. 2, see Supp. Inf., Sec. H. Although these peaks suggest that semantic encoding leverages a higher-dimensional abstract space—echoing findings from brain-similarity studies (Antonello and Cheng, 2024)—we defer a deeper investigation of representation dimensionality to future work.

As a null hypothesis, we show in Supp. Inf. E that performing batch-shuffling on any of the datasets leads to completely uninformative representations (Information Imbalance gives 1 for every layer), since the semantic correspondence between translations is destroyed. Furthermore, in Supp. Inf. F we show that our results are compatible with those obtained using the overlap neighborhood metric from Huh et al. (2024). In Supp. Inf. D we also show that, if instead of considering many-token representations as we do in Fig. 2, one reduces the dimensionality by averaging all tokens, as done in Huh et al. (2024), the results are qualitatively compatible, but present significant quantitative differences. For all layers, but most strongly on the first and the last layers, we observe that averaging significantly increases the level of mutual predictability.

### 3.1.1 HETEROGENEITY OF DEEPSEEK'S REPRESENTATIONS FOR DIFFERENT LANGUAGE PAIRS

We repeated the previous translation experiment with other language pairs, in particular English-Italian, English-Spanish, Italian-Spanish and German-Hungarian. We considered again sentences taken from opus books (Tiedemann, 2012) and applied the same filtering procedure regarding their number of tokens, as in Fig. 2a).

Fig. 2b) shows the same qualitative behavior observed for English-Spanish pairs (low Information Imbalance in the deep layers) but with major quantitative differences. Even if the training dataset of DeepSeek-V3 contains several languages in which the network can perform well, the quality of the representations is heterogeneous across languages, since it depends on the amount of data seen in each language. Fig. 2b) shows that the Information Imbalance between English and Italian is higher than between English and Spanish. A possible interpretation is that models may encode richer Spanish representations because Spanish appears much more frequently than Italian in large web corpora—Common Crawl, for instance, contains roughly twice as many Spanish as Italian documents (Lan, 2025; Common Crawl Foundation, 2025). Aligned with this argument, we observe that, even though Spanish and Italian are linguistically closer than either is to English, the Information Imbalance between Spanish and Italian is higher than between English-Spanish or English-Italian translation pairs. Finally, we observe that translations involving a less common language, namely Hungarian, are associated with a significantly higher Information Imbalance (the minimum value is 0.5).

### 3.1.2 SEMANTIC INFORMATION IN LLMS IS SPREAD ACROSS MANY TOKENS

Having shown that long-span Information Imbalance captures the semantic similarity of sentences in different languages, we turn our attention to the extent to which different tokens are contributing to cross-linguistic similarity. Fig. 3a) shows the minimum Information Imbalance across all layers as a function of the number of tokens involved in the calculation, again for English-Spanish translations from opus books. The span is determined starting from the end and moving back towards the sentence beginning. For both models, we see that concatenations of tokens increases mutual predictivity: this suggests that layer summaries – for example the mean or the last tokens– do not necessarily capture all the semantic information. In facts, if we take alignment as a proxy of semantic quality, this suggests that semantic information is not concentrated in the last tokens, but spread over many of them.

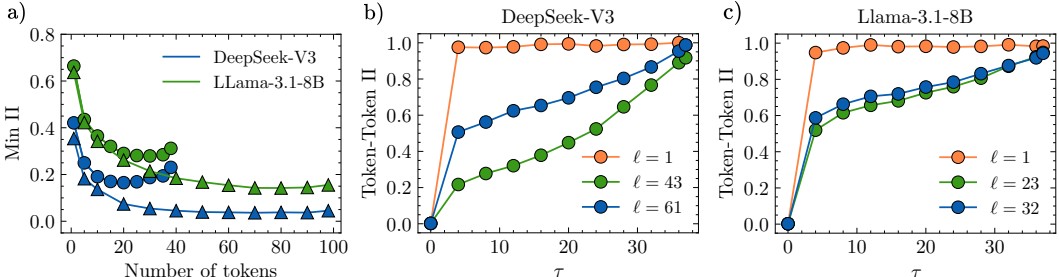

Figure 3: **Panel a)** Minimum Information Imbalance across depth, from English to Spanish representations, as a function of the number of tokens used in the computation, for DeepSeek-V3 and Llama3.1-8b, when considering shorter (40 to 80 tokens, in circles) or longer (100 to 200 tokens, in triangles) sentences. **Panels b) and c)** Information Imbalance from the last token to a previous token at token-distance $\tau$, using English sentences, computed on the representations generated by DeepSeek-V3 in panel a) and by Llama3.1-8b in panel b), in different layers. Long-distance token-token correlations are maximal (Information Imbalance increases most slowly) for representations on the inner semantic layers (43 for DeepSeek-V3, 23 for Llama3.1-8b), and the effect is dramatically stronger in DeepSeek-V3. The Jackknife error bars (five repetitions subsampling 2,000 of 5,000 sentences) are too small to show; the marker sizes act as an upper bound on these errors.

Focusing first on DeepSeek-V3, we note that, up to about 10 tokens, the results do not change much between shorter (40 to 80 tokens) and longer (100 to 200 tokens) sentences. In both cases adding more tokens significantly boosts the information content. However, for shorter sentences we observe that the Information Imbalance reaches its lowest value at around 20 tokens, and it increases with the addition of further tokens. We conjecture that this is because the last added tokens are close to the start of the sentence, and thus they do not carry much information about the semantic content of the whole sentence, given the autoregressive nature of the attention mechanism in the considered language models. In contrast, for longer sentences, we observe that adding tokens leads to a rapid decrease in Information Imbalance, which converges to very low values (around 0.05) at around 50 tokens and stays there. On the one hand, this reflects the fact that, as longer sentences contain more information, their translation will contain more shared information at the semantic level. On the other hand, it further corroborates that information is spread across a large number of tokens. We observe similar trends for Llama3.1-8b Information Imbalance, although the curves are systematically above the DeepSeek-V3 ones. Similar to what we observed in Fig 2a), this confirms that bigger and better performing models generate representations that align better and, potentially, of better quality, capturing deep semantic similarity across different languages. This result shows that not every token contributes homogeneously to the semantic similarity between representations, and thus this analysis suggests to take the concatenation of roughly the last 20 tokens as representative of the sentence content.

### 3.1.3 INNER REPRESENTATIONS HAVE LONGER-RANGE CORRELATIONS

The previous analysis shows that considering several tokens improves alignment across different languages One wonders if the activations of the tokens in those layers are uncorrelated or not. If activations were independent, one could argue that different tokens would be associated to different "directions" in semantic space. If they are correlated, this would suggest that the model is building a joint semantic representation of sentences spreading its information across its component tokens. We address this question by measuring how informative two distinct tokens of the same sentence are about each other. Fig. 3 (panels b) and c)) shows how much the Information Imbalance between two token positions depends on their distance, for sentences in English. Concretely, we measure $\Delta(T \rightarrow T - \tau)$, the Information Imbalance between the token at position $T$, and tokens at position $T - \tau$, for $\tau = 1, 2, \ldots$. We report this measure for the first and last layers, as well as an intermediate layer from the semantic region (layers 43 of DeepSeek-V3 and 23 of Llama3.1-8b, respectively). For both LLMs, the Information Imbalance rapidly grows as a function of $\tau$ for the initial layers and saturates to 1, clearly showing that the first layer is short-range correlated. For the deeper layers that

were shown in Fig. 2 to associate to the phase of shared cross-linguistic information, the Information Imbalance exhibits the slowest growth, meaning that far away tokens are predictive about each other, i.e., in these layers the information is mostly shared between tokens. DeepSeek's deep representations have notably lower Information Imbalance across tokens than Llama3.1-8b's, hinting at a connection between (i) the quality of the semantic representations in terms of token alignments and (ii) the magnitude and the span of long-distance correlations. A clear separation in effect strength between the intermediate semantic layer and the last layer (61 for DeepSeek-V3, 32 for Llama3.1-8b) is moreover only present for DeepSeek-V3. Still, even for DeepSeek-V3 the last layer has Information Imbalance values clearly below those of the first layer, suggesting that putatively, while the last layer is optimized to write the output, it still carries semantic content. To investigate how semantic information is distributed across tokens and whether causal asymmetries differ across languages, we conduct a detailed token-level analysis reported in Appendix I.

## 3.2 IMAGES SHARING HIGH LEVEL SEMANTICS

We extend the analysis to pairs of images and measure how vision transformers (ViTs) capture semantic information for two systems: image-gpt (Chen et al., 2020) and DinoV2 (Oquab et al., 2024). Image-gpt performs next-token prediction after it (i) down-scales the images; (ii) unrolls them, juxtaposing the rows; (iii) quantizes the colors. It mimics the strategy of LLMs: the last layers must perform next-token prediction; previous work suggests that semantics appears in the middle Chen et al. (2020). DinoV2 encodes salient features. It trains a student network to mimic its teacher and reconstruct different instances of the same image. The final layer should be the most semantically rich: its output is the input to downstream tasks such as depth estimation, image segmentation, and instance retrieval.

We first compare representations for pairs of images from the Imagenet1k dataset (Deng et al., 2009). Each couple shares the same class, which we take as a proxy of similar semantics: we process the pairs of instances with the same architecture and compute the Information Imbalance on a random subset of 2500 same-class pairs of images, which we sample without repetition. We replicate the procedure five times to quantify uncertainty. We report the results in the left panel of Fig. 4. As the assignment of same-pair images to two spaces used to compute the Information Imbalance is arbitrary, we report the average of Information Imbalance in the two directions. The minimum of the Information Imbalance is in the middle for image-gpt and at the end for DinoV2, consistently with previous work (Valeriani et al., 2023). The Information Imbalance is lower for DinoV2 ($\approx 0.43$): this would correspond to roughly 10% shared features in the Gaussian case, see Fig. 1.

## 3.3 IMAGES AND CAPTIONS: MULTIMODAL DATA SHARING SEMANTIC CONTENT

Image-caption pairs encode the same semantic content in different modalities: we are interested in their representations as semantics must play a key role in alignment. We process text with DeepSeek-V3 and images with DinoV2 and image-gpt. The dataset is flickr30k (Young et al., 2014): it contains pairs of images and captions; the captions are the concatenations of five different human-generated descriptions. We next repeat the analysis of Fig. 3a), estimating the minimum Information Imbalance between the image transformer and DeepSeek-V3 as a function of the number of image tokens concatenated from the end of the sequence. For DeepSeekV3, we fix the token window to 9, which corresponds to the point where the steep decrease of Information Imbalance observed in Fig. 3a) essentially saturates. We find that, for both DinoV2 and image-gpt, it is necessary to concatenate the last 200 tokens to capture the relevant shared information. Results obtained with shorter windows (last 50 tokens) and with mean-token are reported for comparison in Supp. Inf., Sec. K.

In Fig. 4 we report the Information Imbalance between layer 52 of DeepSeek-V3 (captions), DinoV2 and image-gpt (images). We select layer 52 of DeepSeek-V3 because it is a layer in which the representations of text in different languages is highly mutually predictive, see Fig. 2. We recover the exact same semantic regions provided by the analysis of the image-image pairs: towards the end for DinoV2 and in the middle for image-gpt. DinoV2 has a lower Information Imbalance of 0.4. Using the Gaussian process as a reference (see Fig. 1), this would correspond to approximately 10% of shared features. The minimum Information Imbalance for image-gpt is $\approx 0.6$, corresponding to 5% of shared features. The dashed lines in Fig. 4 report the Information Imbalance from the image to the text representation. Remarkably, for the image-gpt model the Information Imbalance

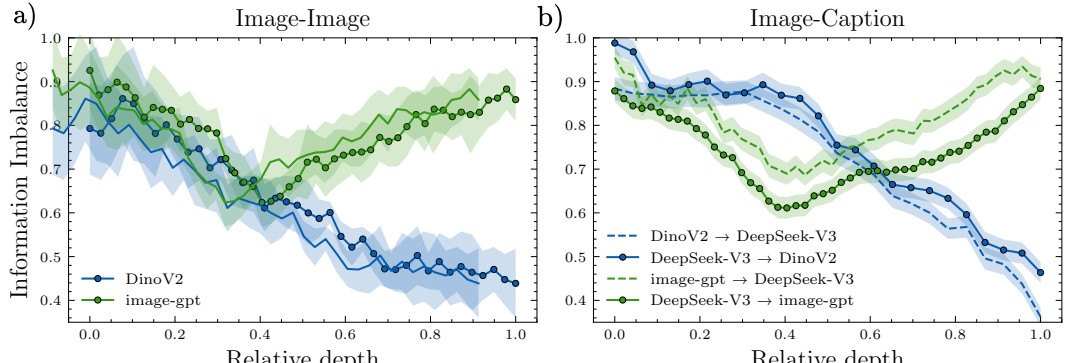

Figure 4: **Panel a)** Information Imbalance for image pairs from the Imagenet1k dataset. We sample 2500 pairs of images from the same class at random; averaging over five replications. We report the mean and standard deviation. **Panel b)** Information Imbalance between image and caption representations on the flickr30k dataset. Images are encoded using DinoV2 and image-gpt-large, while captions are processed with DeepSeek-V3. We report the imbalance as a function of the relative depth in the vision transformer, using the 52nd layer of DeepSeek-V3 (inside the semantic region) for captions. Results are averaged over 5000 samples, with uncertainty estimated via bootstrapping (100 replicas of 200 samples each). Dotted lines indicate information flow from caption to image; dashed lines, from image to caption.

is considerably larger than in the text-to-image direction. In particular, the minimum value of the Information Imbalance is $\approx 0.7$. The difference between the Information Imbalance in the two direction is significantly higher than the error bars estimated by bootstrap, indicating that the effect is statistically significant. This difference is likely due to the better quality of the DeepSeek-V3 representations. The asymmetry in the Information Imbalance is reversed for the DinoV2 model. In this case the image representation predicts the text representation marginally better than the reverse. This can be due to the fact that DinoV2 is explicitly trained to extract semantically meaningful visual features, and indeed the largest asymmetry is observed in the last layer.

To directly test whether joint text–image training facilitates cross-modal alignment, we performed a set of experiments with the ViT-L/14 visual encoder of the multimodal CLIP system (Radford et al., 2021) on the same flickr30k dataset (Supp. Inf., Sec. K). We computed the Information Imbalance between CLIP image representations and DeepSeek-V3 text representations and compared it with the DinoV2–DeepSeek-V3 baseline. We found that the DeepSeekV3-CLIP Image Information Imbalance reaches a minimum of approximately 0.3 ($\approx 15\%$ of shared features), which is lower than the minimum Information Imbalance observed for DinoV3-DeepSeeV3, indicating that CLIP's visual encoder aligns more closely to the text semantics than a purely visual model of comparable capacity.

## 4 LIMITATIONS

It would be interesting to extend our analyses to a further variety of models of intermediate sizes and diverse architectures, including diffusion models, for example quantifying the depth and the length of the semantic regions found as a function of the number of parameters in the model, and to consider models that specifically target multilinguality (Martins et al., 2025; Üstün et al., 2024). In our experiments with translations, we used human translations from opus books (Tiedemann, 2012), in which the different language pairs may correspond to translations from different novels, introducing an extra source of variation not taken into account in our analysis. We explicitly avoided using LLMs to generate the translations to avoid the introduction of biases in the analyses. Indeed, we found that, for the case of English-Italian, DeepSeek-generated translations have more similar representations than those obtained from opus books. We worked with a limited number of language pairs, that allowed us to spot interesting heterogeneities across representations, and to propose some plausible interpretations on why they appear. We leave for future work a detailed quantitative analysis solely dedicated to the influence of language heterogeneity in the mutual information

content of representations. For example, fixing English as a pivot language and then systematically considering different translations of the same English text into languages with different online presence, and of different degree of relatedness. Finally, it would also be important to extend our analysis to embedding models trained specifically for cross-lingual retrieval, in order to assess whether architectures optimized for cross-lingual alignment exhibit different patterns of semantic convergence than general-purpose LLMs.

## 5 CONCLUSION

Through a novel application of the Information Imbalance measure (Glielmo et al., 2022), we showed that DeepSeek-V3, the largest publicly available LLM, developed an internal processing phase in which different inputs that share the same semantics, such as translations and image-caption pairs, are reflected in representations that are extremely similar. We further ascertained that, on these "semantic" layers, long token spans meaningfully contribute to the representation. Several of the patterns we observed in DeepSeek-V3 also emerge in the medium-sized Llama3.1-8b model, but in a weaker manner, suggesting that deep semantic processing is a hallmark of better models. We also analyzed visual processing models, finding again the presence of layers that capture deeper semantic similarity, and whose position depends on the objective the models are trained upon. DeepSeek-V3 textual representations and vision transformer visual representations of the same concepts are most strongly aligned in the respective semantic layers.

Our work supports the hypothesis that, as deep models improve, they converge towards shared representations of the world (Huh et al., 2024). We took a first step away from simply verifying that such representations exist, towards characterizing their nature, by precisely quantifying the degree of shared information, localizing where they occur in different networks and determining how they are synthesized from composite inputs. Future research, besides extending the empirical coverage of our investigation, should further increase the resolution of the measurements, allowing us to go from observing the holistic behavior of a model with respect to a data set to a full characterization of how semantic representations are constructed given each specific input instance.

## 6 ETHICS STATEMENT

This work adheres to the ICLR Code of Ethics. Our experiments employ publicly available datasets and open-source models (under their respective licenses). We are not aware of any ethical concerns. To the contrary, by providing new tools to understand the inner workings of LLMs and other AI systems, we hope to contribute to a safer and more transparent AI.

## 7 REPRODUCIBILITY STATEMENTS

We describe all model configurations, datasets, and computational settings in the main text and supplementary material to enable independent verification. Upon acceptance, we will release the full source code and instructions in a public repository to facilitate reproducibility.

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

# Supplementary Information

## A COMPUTE RESOURCES

We run DeepSeek-V3 on a cluster of 16 H100 GPUs (80GB each), using the SGLang framework (Zheng et al., 2024). For LLama3.1-8B, a single GPU from a DGX H100 system is sufficient. For our experiments on text and image-caption pairs, 1TB of RAM is enough, moving to the GPU a couple of layers at a time. Image-Image experiments are done using up to five NVIDIA A30 GPUs and 500 GB of RAM.

## B ASSETS

opus books: https://huggingface.co/datasets/Helsinki-NLP/opus_books ; licence: CC BY 4.0.

DeepSeek-V3: https://huggingface.co/deepseek-ai/DeepSeek-V3 ; Model licence.

Llama3.2-1b: https://huggingface.co/meta-llama/Llama-3.2-1B ; license: llama3.2

Llama3.2-3b: https://huggingface.co/meta-llama/Llama-3.2-3B ; license: llama 3.2

Llama3.1-8b: https://huggingface.co/meta-llama/Llama3.1-8b-3.1-8B ; license: llama3.1

BERT: https://huggingface.co/google-bert/bert-base-multilingual-cased ; licence: apache-2.0

Flickr dataset: https://huggingface.co/datasets/clip-benchmark/wds_flickr30k ; subject to flickr terms of use.

Imagenet: https://image-net.org/ ; terms of access: https://image-net.org/download.php

DinoV2: https://github.com/facebookresearch/dinov2/tree/main ; licence: Apache 2.0

image-gpt-large: https://github.com/openai/image-gpt ; licence: Modified MIT

## C TRANSLATIONS DATASET

We filter pairs of translated sentences from Helsinki-NLP/opus_books (Tiedemann, 2012) that have a number of tokens between $40$ and $80$, avoiding trivial sentences and incorrect translations, while keeping roughly 5000 thousands sentences of comparable length. We avoid truncating the translated sentences, preserving their full meaning. We excluded the last two tokens of each sentence from the analysis, as they consistently correspond to punctuation marks (e.g., a period or a period followed by a quotation mark), which introduce trivial similarities in the representations.

## D AVERAGE TOKEN COMPARISON

Fig. 5 shows the Information Imbalance between DeepSeek-V3 representations of English and Spanish translations, using 20 tokens of each sentence, with and without taking the average over the token axis. Note that for the case of concatenated tokens, the curve is the same as in Fig. 2 of the main article, for direct comparison. When taking the average, we use the $L_2$ distance and we clip and normalize the inputs, as Huh et al. (2024). When using all 20 tokens we binarize activations and we use the Hamming distance, as in the main text. While the results are qualitatively consistent, they differ significantly in quantitative terms. Across all layers—most notably in the first and last—averaging leads to a marked increase in mutual predictability (lower Information Imbalance), due to the elimination of positional information, which is most relevant in the initial and last layers, i.e., before and after the semantic region.

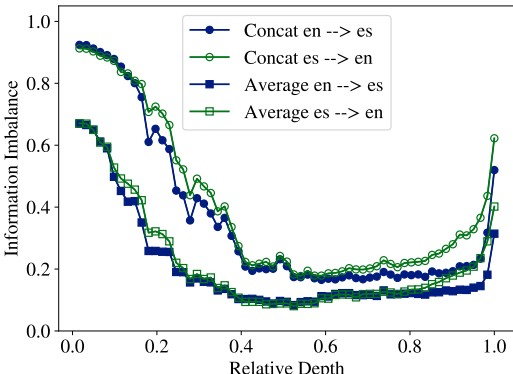

Figure 5: Information Imbalance between English (en) and Spanish (es) representations generated by DeepSeek-V3's as a function of depth. 'Concat' stands for the results obtain with the concatenation of the 20 last tokens of each sentence. 'Average' stands for the average of the same 20 tokens. The standard deviation is computed with a Jackknife procedure, subsampling 2000 samples out of 5000 five times, and it is smaller than marker size.

## E MISALIGNMENT OF TRANSLATIONS ERASES SEMANTIC SIMILARITY

As a consistency check, Fig 6 shows the Information Imbalance for DeepSeek-V3's and Llama3.1-8b representations using misaligned translations, namely performing a batch-shuffle in one of the datasets. Since the semantic correspondence between sentences is destroyed, the representations are not informative about each other, and thus the Information Imbalance is close to one, for all layers. The same occurs with misaligned image pairs, Information Imbalance being around one when pairs do not share the same class, for all layers.

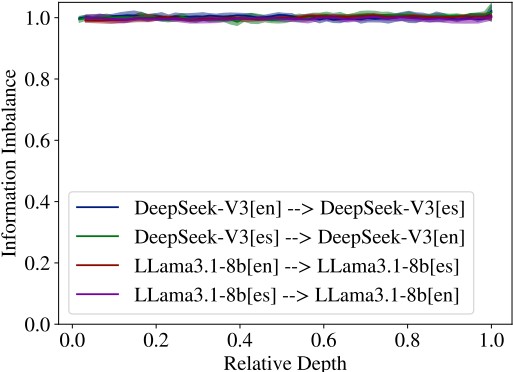

Figure 6: Information Imbalance between English (en) and Spanish (es) representations generated by DeepSeek-V3 and Llama3.1-8b, for the "non-informative case", namely a misaligned dataset in which we batch-shuffle the Spanish translations. The (hardly visible) shaded area corresponds to one standard deviation, computed with a Jackknife procedure subsampling 2000 samples out of 5000.

## F COMPARISON BETWEEN NEIGHBORHOOD OVERLAP AND INFORMATION IMBALANCE

To quantify representation similarity, Huh et al. (2024) used the neighborhood overlap, namely the average fraction of shared $k$ nearest neighbors. In particular, they found this metric to be more suitable than linear metrics, like Central Kernel Alignment (CKA), providing a stronger signal. In this section we compare our results from Fig. 1.a) of the main article with the neighborhood overlap computed on the same representations. To measure the neighborhood overlap we follow the pipeline

of Huh et al. (2024), namely we clip the activations using quantiles of order $0.05\%$ and $95\%$, we normalize each vector, and we measure the distances between them with the $L_2$ metric. Fig 7 shows that the neighborhood overlap between representations increases and reaches a plateau concurrently with the local minimum plateau of the Information Imbalance, thus both methods give qualitative the same results. We note that the neighborhood overlap, being defined between 0 and 1, here only reaches a value around 0.2. Similar small values were also observed and even highlighted in Huh et al. (2024), although the authors do not provide arguments to understand the reasons behind this property. We note that if the $k$-th neighbor of representation $A$ is instead the $k+1$ neighbor of representation $B$, automatically that point counts as outside the $k$-neighborhood, even if it is extremely close, rendering the alignment value very low. This effect doesn't take place in the Information Imbalance, which, instead of evaluating if $k$ neighbors are shared, measures what is the rank in space $B$ of $k$ neighbors in space $A$, and vice-versa.

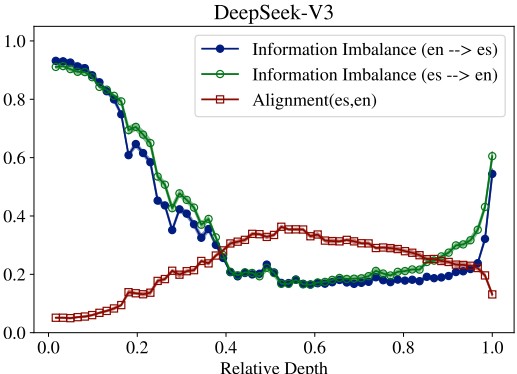

Figure 7: Information Imbalance between English (en) and Spanish (es) representations generated by DeepSeek-V3 from Fig. 2 of the main article, compared with the neighborhood alignment metric from Huh et al. (2024). The (hardly visible) shaded area corresponds to one standard deviation, computed with a Jackknife procedure subsampling 2000 samples out of 5000.

## G  BINARIZING HIGH DIMENSIONAL REPRESENTATIONS HAS A MARGINAL EFFECT ON THE INFORMATION IMBALANCE

Fig. 8 includes again, as reference, the Information Imbalance between DeepSeek-V3 representations of English and Spanish translations, using the last 20 tokens of each sentence, binarizing the activations and using the Hamming distance, (results of Fig. 2 from main text). To investigate the effects of binarizing the representations, we show the Information Imbalance using the same data where, instead of taking the sign of the activations to generate binary variables, we clip them using quantiles of order $0.05\%$ and $95\%$, we normalize vectors to unit norm, and we use the $L_2$ distance, similar to Huh et al. (2024). Remarkably, binarizing has marginal effects on the Information Imbalance computed with full precision (for DeepSeek-V3's representations, BF16), which is possible in this setup given that we are working with 20 tokens.

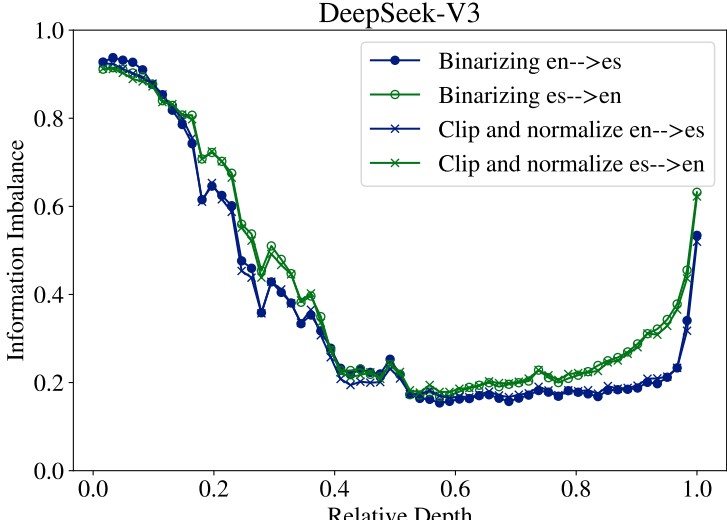

Figure 8: In circles, Information Imbalance between English (en) and Spanish (es) representations generated by DeepSeek-V3, binarizing the activations and using the Hamming distance. In crosses, the Information Imbalance clipping and normalizing activations, and using the $L_2$ metric. The standard deviation, computed with a Jackknife procedure subsampling 2000 samples out of 5000, is smaller than marker size.

## H  BINARY INTRINSIC DIMENSION OF DEEPSEEK-V3 REPRESENTATIONS

The Binary Intrinsic Dimension (BID) is defined in Acevedo et al. (2025) through the probability distribution of Hamming distances between samples of binary high dimensional vectors. If we consider $N$-dimensional vectors $\boldsymbol{\sigma}$ with components $\sigma_i$, $i = 1, ..., N$ uniformly distributed in $\{0, 1\}$, then the probability of observing Hamming distance $r$ between any two samples is exactly $P_0(r) = \frac{1}{2^N} \binom{N}{r}$. The BID is defined as the coefficient $d_0$ of the following Ansatz:

$$P(r) = \frac{1}{2^{(d_0 + d_1 r)}} \binom{N}{d_0 + d_1 r},$$  (1)

where $d_1$ is a second variational parameter. We perform a second-order optimization of the Kullback-Leibler divergence between the model equation 1 and the empirical distribution of distances, computing numerical derivatives with JAX's autodiff. For more details on the BID computation, see Acevedo et al. (2025). We used hyperparameters $\alpha_{min} = 0.15$, and $\alpha_{max} = 0.4$. Fig. 9 shows the Binary Intrinsic Dimension (BID) of DeepSeek-V3's representations processing English sentences from opus books with length between 40 and 80 tokens. We take the last 10 and the last 20 tokens of the binarized representation, and we find two dimensionality peaks that roughly coincide with the beginning (around 0.4) and the end (around 0.9) of the semantic region found in Fig. 2. These results align with recent evidence of abstract spaces presenting dimensionality peaks in deep representations (Cheng et al., 2025; Antonello and Cheng, 2024). The missing points at the beginning of the blue curve correspond to failed optimizations due to the strong multimodality of the empirical distribution of distances.

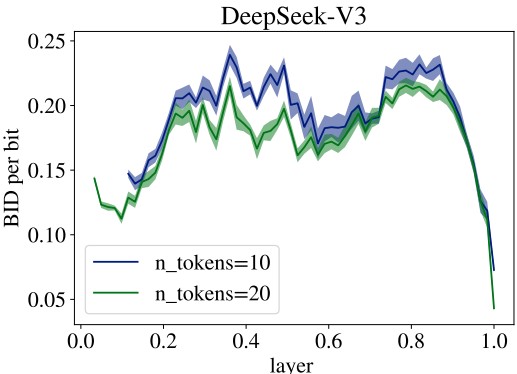

Figure 9: Binary Intrinsic Dimension (BID) normalized by the number of bits (neurons), in the representation generated by DeepSeek-V3 processing English text. In the legend, n_tokens stands for the number of tokens concatenated in the representation, starting from the last. Shaded area corresponds to one standard deviation, computed with a Jackknife procedure, subsampling 2000 samples out of 5000, ten times.

# I  TOKEN-LEVEL INFORMATION IMBALANCE AND CAUSAL ASYMMETRY ACROSS LANGUAGES

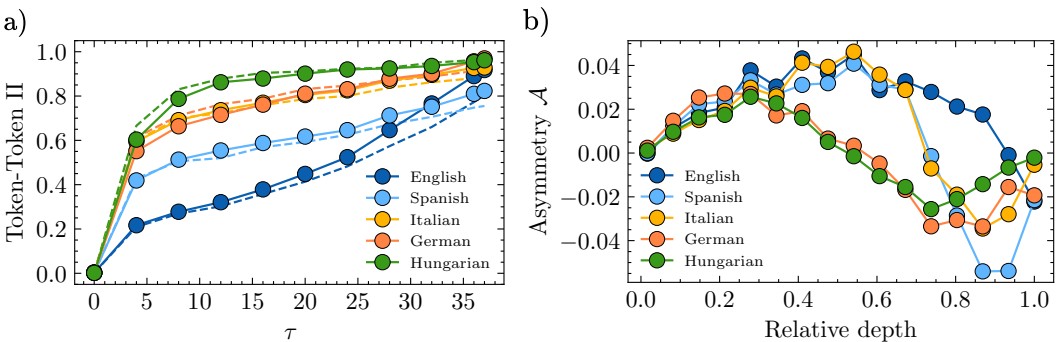

Figure 10: **Panel a)** In solid markers, the "backward" Information Imbalance, i.e., from the last token to a previous token at token-distance $\tau$, as a function of $\tau$, for DeepSeek-V3 representations at layer 43 and sentences in several languages. In dashed lines, the reciprocal "forward" Information Imbalance. **Panel b)** Asymmetry $\mathcal{A}$ between forward and backward Information Imbalance, averaged over $\tau$, as a function of DeepSeek's relative depth. The Jackknife error bars (five repetitions subsampling 1,000 of 2,000 sentences) are too small to show; the marker sizes act as an upper bound on these errors.

Fig. 10a) focuses on a semantic layer from DeepSeek-V3, and it shows token-token Information Imbalance as a function of token distance for several languages. We present the Information Imbalance in both directions: The "backward" Information Imbalance, from the last token, $T$, to a previous token at distance $\tau$, $\Delta(T \to T-\tau)$, and the reciprocal or "forward" Information Imbalance, $\Delta(T-\tau \to T)$. We observe that English deep representations are much more correlated than the representations in the other languages. We conjecture that this is a sign of better quality, consistent with Fig. 2b), where the presence of English in a pair results in the best predictability (lowest Information Imbalance scores). At the opposite extreme, note that single-language German and Hungarian representations are the least correlated in Fig. 10a), and the German-Hungarian translations have indeed the worst mutual predictability score (highest Information Imbalance) in Fig. 2. Furthermore, Spanish and Italian have the second and third most correlated inner representations in Fig. 10a), coherently with the profiles of the English-Spanish and English-Italian pairs in Fig. 2b) (lowest and second lowest Information Imbalance values, respectively).

Finally, Fig. 10a) shows that, for English, as the distance between tokens $\tau$ increases, the relative information content between past and future tokens gets increasingly asymmetric. In particular, earlier tokens are more informative of the last one than the other way around. This could be expected, since causal models like DeepSeek-V3 are trained to predict future tokens, and text itself has a natural causal asymmetry. Nonetheless, Fig. 10a) shows that this past-to-future asymmetry is heterogeneous across languages. To further study the effect, we first define the *asymmetry* as $\mathcal{A} = \Delta(T \rightarrow T - \tau) - \Delta(T - \tau \rightarrow T)$. When $\mathcal{A}$ is positive, the earlier token predicts the last one more than the latter predicts the earlier token, whereas a negative $\mathcal{A}$ has the opposite interpretation. Fig. 10b) shows the information asymmetry $\mathcal{A}$ of DeepSeek-V3 as a function of its layers for the all languages we studied. For initial layers (i.e., relative depth less than roughly $0.4$, see Fig. 2), we observe that all representations have a comparable amount of causal asymmetry. Instead, for semantic layers there are strong heterogeneities between languages. English remains causally asymmetric until the last few layers of the network, while the other languages present earlier negative values of $\mathcal{A}$. Intriguingly, Spanish and Italian, the two languages with the lowest Information Imbalance values from English, have a later dip in $\mathcal{A}$ compared to German and Hungarian. These results call for further experiments to study possible relationships between these observed information asymmetries, model performance, and linguistic structure. We leave them as material for future research. In Sec. J we study this information asymmetry for a non autoregressive model.

## J BERT'S FORWARD-BACKWARD ASYMMETRY PROFILE

In this section we study, the asymmetry in II between two tokens at distance $\tau$, $\mathcal{A} = \Delta(T \rightarrow T - \tau) - \Delta(T - \tau \rightarrow T)$, where $T$ is the last token, to see if the observed asymmetries are similar to those of DeepSeek-V3 (Fig. 10b). Fig. 11 shows the asymmetry $\mathcal{A}$ for similar English sentences from opus books as those used in Fig. 10b), processed by BERT. Contrary to what was observed in Fig. 10b), BERT's representations present a positive asymmetry close to the initial and final layers, and negative asymmetry in central layers. This strong difference of asymmetry profiles opens the door for further studies on the asymmetry of correlations in several different models and architectures, and its potential relations to downstream tasks.

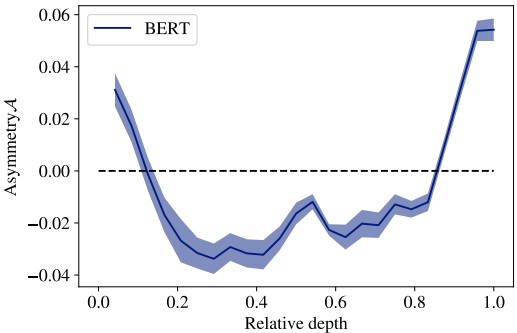

Figure 11: Asymmetry $\mathcal{A}$ between forward and backward Information Imbalance, averaged over token all distances as a function of BERT's relative depth. The shaded area corresponds to one standard deviation computed subsampling half out of 7500 samples, five times.

## K IMAGE-CAPTIONS PAIRS

In Fig. 12, we report additional results for the image-caption experiments: we consider different windows of tokens, using the last 200 tokens for images –as we do in the main text– vs a smaller windows of 50 tokens and the average. In general, results are qualitatively similar across setups. We observe that the choice of the number of tokens comes with a trade-off: more tokens convey more information; at the same time, having more tokens implies working with bigger objects.

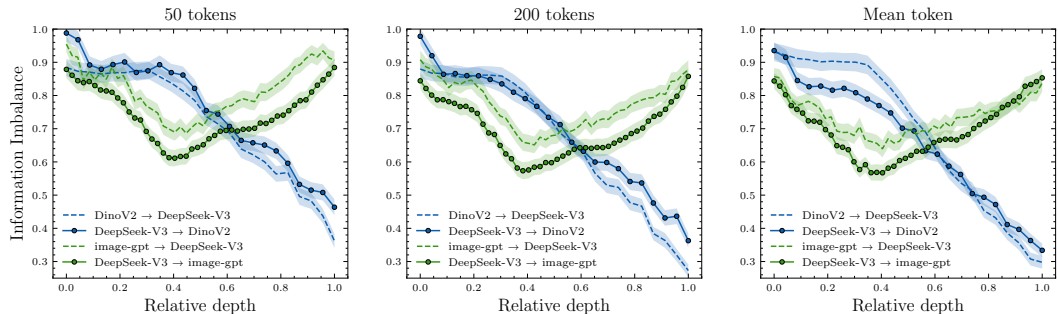

Figure 12: Information Imbalance for image-caption pairs. We consider three scenarios: (i) 50 tokens; (ii) 200 tokens; and (iii) the mean token. The results are qualitatively similar.

In Fig. 13, we show how the alignment between models changes when the number of tokens is increased: on the left, we report the case for DinoV2, comparing it with DeepSeek-V3, in the image-caption case, and with itself for the pairs of images. On the right, we report the same analysis for image-gpt.

Finally, we performed three analyses on the flickr30k image–caption pairs to probe how CLIP's joint training influences representational alignment.

1. DinoV2 → CLIP-Image: comparing a purely visual model with CLIP's visual branch.

2. DeepSeek-V3 → CLIP-Text: comparing a purely textual model with CLIP's text branch.

3. DeepSeek-V3 → CLIP-Image: comparing a purely textual model with CLIP's visual branch.

Results are in Fig. 14. In all cases we compute the Information Imbalance in both directions (from the source model to the target and vice-versa), always indicating the direction in the figure legends. The DeepSeek-V3 → CLIP-Image comparison directly tests whether a visual encoder trained to align with text predicts textual representations better than a unimodal vision model. Its minimum Information Imbalance is approximately 0.3 ($\approx 15\%$ shared features), which is lower than the $\approx 0.40$ minimum for DeepSeek-V3 → DinoV2 (main text Fig. 4). The DinoV2 → CLIP-Image Information Imbalance remains extremely low ($\approx 0.01$) past a relative depth of $\approx 0.8$, consistent with a near-identical visual representation. Conversely, the DeepSeek-V3 → CLIP-Text Information Imbalance decreases more gradually and stabilizes around 0.4, indicating weaker alignment of CLIP's text branch to purely textual models. Shaded regions in Fig. 14 show 95% bootstrap confidence intervals.

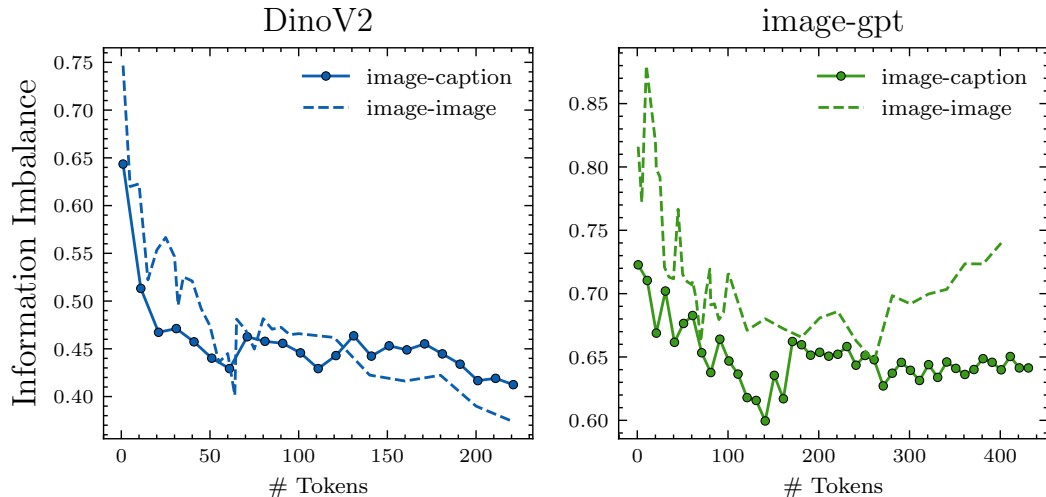

Figure 13: Information Imbalance for varying windows of tokens. We report the minimum of the Information Imbalance, a proxy of alignment, for growing windows of tokens. On the left, we report the results for DinoV2 in both the image-caption (dotted blue line) and the image-image case (dashed blue line). On the right, we report the results for the image-gpt in both the image-caption (dotted green line) and the image-image cases (dashed green line).

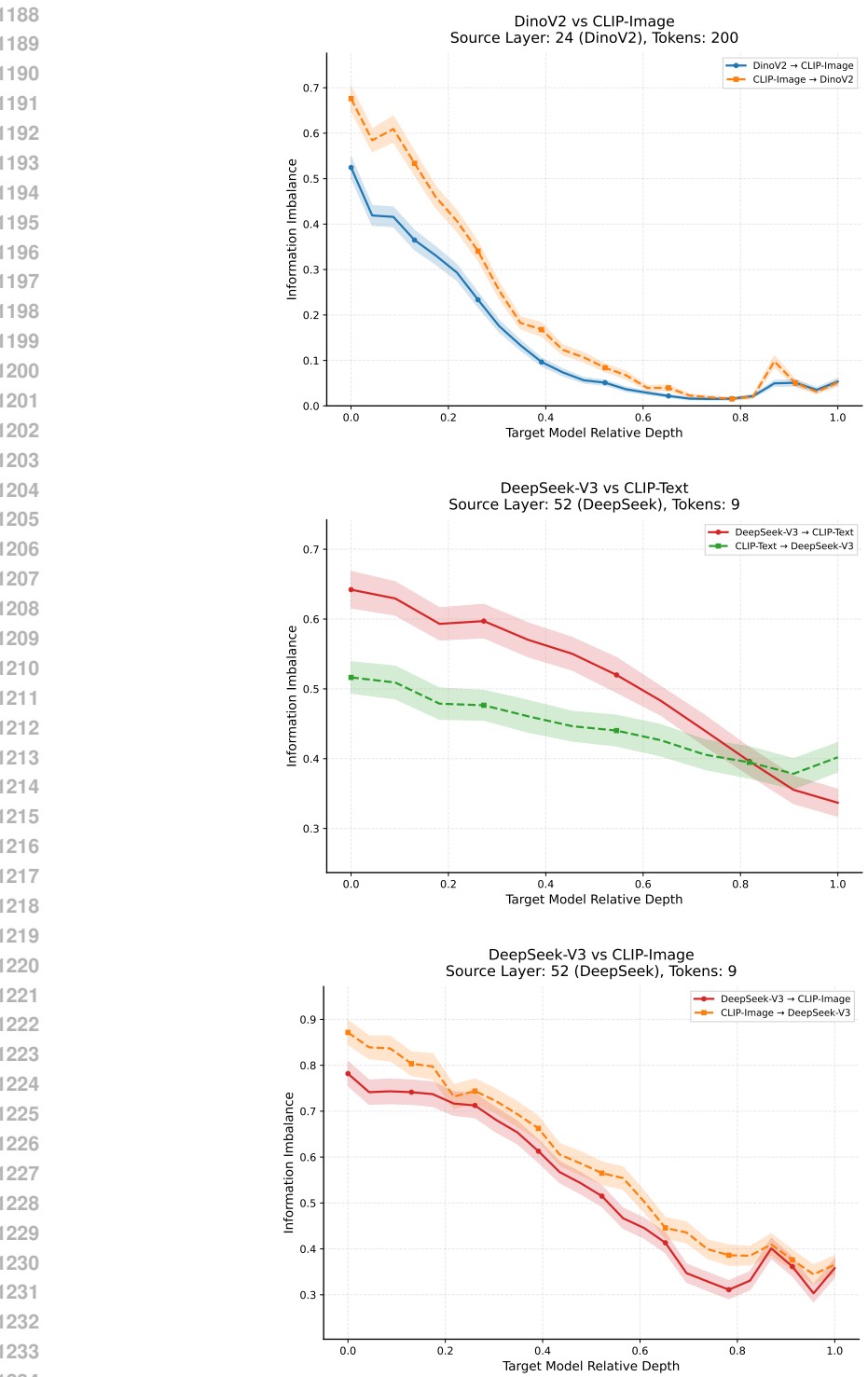

Figure 14: Information Imbalance on flickr30k involving CLIP representations. Top: DinoV2 image vs. CLIP image. Middle: DeepSeek-V3 text (layer 52) vs. CLIP text. Bottom: DeepSeek-V3 text vs. CLIP image (new cross-modal analysis). Lower Information Imbalance indicates stronger semantic predictivity. Shaded areas show 95% bootstrap confidence intervals.

