# OpenReview forum: "A quantitative analysis of semantic information in deep representations of text and images"
_ICLR.cc/2026/Conference — Submitted to ICLR 2026_

### Official Review · Reviewer_GtuN · 2025-10-30

**Soundness:** 2
**Presentation:** 3
**Contribution:** 2
**Rating:** 4
**Confidence:** 3

**Summary:**

This paper proposes Information Imbalance (II), a directional and asymmetric metric designed to assess the semantic alignment between representations across languages, modalities, layers, and tokens. The authors utilize II to identify “semantic layers” in both language and vision models, revealing how semantic information is distributed across layers and tokens. The study is empirically grounded on translation pairs from OPUS Books and image-caption pairs from Flickr30k, comparing representations from DeepSeek-V3 (a large mixture-of-experts model) with those from LLaMA-3.1-8B, as well as from DinoV2 and image-GPT.

This paper presents a clearly structured and methodologically consistent investigation of semantic structure in large models. The use of Information Imbalance as a unifying probe is appealing and the findings are generally well-motivated. However, the experimental scope is currently too narrow to support some of the broader claims, and the dependence on a single metric and representation choice reduces the robustness of the conclusions.

**Strengths:**

1. The topic is timely and of broad relevance to the community, particularly as interpretability and quantitative understanding of large models become increasingly central. In particular, the paper is motivated by a fundamental and well-posed question: where, how, and to what extent semantic information is encoded within and across model modalities.

2. The analysis confirms and consolidates several important intuitions about semantic structure in large models, including the presence of mid-to-late semantic layers, asymmetries in token dependencies, and layerwise cross-modal alignment.

**Weaknesses:**

1. The paper’s central question is ambitious, but the empirical evidence is too limited to support it.
The work aims to explain where and how semantic information is encoded across models and modalities, yet the experiments cover only two language models (DeepSeek-V3 (MoE, 671B) and LLaMA-3.1-8B (dense, 8B)) and a single multimodal dataset (Flickr30k). Without intermediate scales, architecture controls, or broader datasets, the conclusions about model size, semantic bands, and cross-modal generality remain under-supported and potentially confounded.

2. The definition of “semantic information” is overly tied to a single metric (II) and representation choice. The operationalization of II relies on binarized activations and small-k neighbor overlap, which may neglect richer structural information such as real-valued distances or rank-order consistency. Although the supplemental material includes one real-valued comparison, the overall analysis lacks rigorous metric ablation or validation using rank-sensitive alternatives such as NDCG or Kendall’s tau. Overall the choice can be acceptable, but it needs side-by-side validation to acknowledge the method’s limits.

3. Several findings largely reiterate existing observations, and the related work review is insufficiently comprehensive.
Prior research has already shown that mid-to-late layers encode more transferable and semantically structured features, that larger models exhibit stronger long-range token dependencies, and that cross-modal alignment often emerges at specific intermediate layers. While applying the II metric as a probe for information content is novel and conceptually interesting, the overall advancement over these established insights remains limited. Moreover, the discussion of related work does not sufficiently situate the paper within this literature, making it difficult to assess the true scope of novelty. A more thorough review and positioning of prior probing and representation studies would help clarify the paper’s contribution.

**Questions:**

1. The comparison between DeepSeek-V3 and LLaMA-3.1-8B conflates scale with architecture. Have the authors considered evaluating a family of dense models (e.g., LLaMA 7B, 34B, 70B or Qwen 7B to 72B) to assess whether the widening of semantic bands correlates with model size independent of MoE structure?

2. All cross-modal results are derived from Flickr30k. Can the authors evaluate whether the trends in II persist on more diverse and challenging datasets, such as COCO or web-scale datasets like CC3M or CC12M?

3. Can II be linked to downstream performance metrics? For instance, are lower II values predictive of improved retrieval performance (e.g., Recall@k, NDCG) in multilingual or image-text tasks? Can II be used to guide pruning or layer selection for distillation?

4. To what extent do the observed directional asymmetries depend on decoding directionality versus architectural factors? Have the authors tried bi-directional or masked LMs to test whether left-to-right asymmetry persists?

---

> ### Author Response · Authors · 2025-11-19
>
> ## Weaknesses
> ### Weakness 1
> Further comparison with other sizes is indeed interesting, and was not added because of the lack of space, and our focus on the largest available open-source model, DeepSeek-V3-671B. We have also observed that Qwen7b has a very similar behavior to the models present in our manuscript, and we will add it as a further comparison.
> In the revised version of the manuscript we will emphasize that our results hold for several different language and vision models with heterogeneous architectures, on different datasets, and this, in itself, shows that our similarity measurements are meaningful and robust. We will remove the emphasis on model size, which, as correctly pointed out by the referee, is not systematically studied on its own.
>
> ### Weakness 2
> The Information Imbalance is a generalized metric with respect to the neighborhood-overlap used in Ref.[3]. There, the authors showed extensive comparisons with respect to CKA, finding that their rank-based metric is preferable, as we comment in our appendix, and we will clarify further in the revision. Moreover the Information Imbalance was already used successfully in Ref. 17, as we discussed in “Related work”.
>
> ### Weakness 3
> We thank the referee for this critique, and we ask them to please provide specific references that we can add to our manuscript to position it better with respect to the literature.
> To the best of our knowledge, the layer dependence of the cross-modal neighborhood alignment was not studied before, nor its explicit dependence on the number of tokens, or the choice of concatenating or averaging tokens. Moreover, binarizing the activations and finding equivalent neighborhoods was also not explored in any previous work that we are aware of, and it is an important result from a theoretical and from a practical point of view. In the revised version of our manuscript we will further  emphasize these points.
>
> ## Questions
> - **Reply 1.** See reply to Weakness 1.
> - **Reply 2.** Yes, we expect our results to hold, given that they are based on the semantic overlap between multimodal data. These experiments are computationally costly, and we prefer to leave them to further work.
> - **Reply 3.** Our paper focuses on the quantitative study of the alignment of representations, and thus these very interesting but costly experiments were out of scope. We conjecture that the layers with the lowest II are the best layers to do cross-modality retrieval, as suggested by the referee, and also they could be the best layers to perform model stitching.
> - **Reply 4.** We will add the results of an analysis on BERT, in which we observed different directional asymmetries. More precisely, BERT representations have a causal asymmetry both in the first and last layers, and they present no asymmetry in deep layers. These results thus suggest that the asymmetries are architecture dependent, and we are going to add them to our manuscript for completeness.

---

> > ### Author Response · Authors · 2025-11-26
> >
> > **Summary of Revisions**
> >
> > We thank all reviewers for their thoughtful feedback. In the revised manuscript, we have:
> >
> > • **Softened and clarified several claims**, explicitly stating limitations and avoiding over-general interpretations.
> >
> > • **Expanded comparisons across model sizes and architectures**, adding results for LLaMA models of different sizes (1B, 3B, 8B) and for BERT. These additional experiments **validate and confirm our main findings**, showing that the key alignment patterns and asymmetries persist across scales and architectures.
> >
> > • **Clarified key methodological points**, including the relation between Information Imbalance and neighbourhood-based measures, the effect of averaging vs. concatenating tokens, and the consistency between binarised and floating-point representations.
> >
> > • **Improved presentation quality**, fixing citation style, figure notation, and overall clarity. All revisions are highlighted in blue in the updated manuscript.
> >
> > We believe these modifications directly address the reviewers’ critiques and substantially strengthen the work.

---

### Official Review · Reviewer_qzxA · 2025-10-31

**Soundness:** 3
**Presentation:** 2
**Contribution:** 2
**Rating:** 4
**Confidence:** 3

**Summary:**

This paper works to measure where and how semantic information emerges within large models for language and vision. They use information imbalance (II) which is defined as an asymmetric proxy for mutual info. II quantifies how much one representation (like a sentence, its translation, etc) predicts another. The authors identify broad semantic layers where cross linguistic and cross modal alignment peak in DeepseekV3 and Llama. They find that in these layers semantic info is distributed across many tokens, correlated over long ranges and exhibits causal asymmetry. Similar analysis of vision transformers show that there are analogous semantic regions and asymmetries with text and image representations. Overall, this study provides qualitative evidence that large models converge toward shared internal semantic spaces across both languages and modalities.

**Strengths:**

The work finds that layer wise analysis identifies where semantics arise in networks, which reveals that meaning is distributed across many tokens and layers.

The paper tests a variety of models, spanning multilingual LLMs, two vision architectures, and multimodal text image pairs, w/extensive validation and controls like shuffled data

This work also supports that larger models exhibit broader and stronger semantic alignment, empirically supporting the Platonic Representation Hypothesis

**Weaknesses:**

This body of work is rather interesting and supports work like PRH. However, there is limited methodological novelty since the contribution seems to be mainly analytical. II metric is borrowed and not newly developed. So the work’s strength lies in empirical breadth rather than algorithmic innovation.

Further, the II values are not intuitive to understand and some readers may struggle to relate these numbers to concrete semantic similarity or task performance (0.2 ≈ 20% shared features).

One other note is that the observations like casual asymmetry or token correlation are also mainly shown for autoregressive models and translation data. This feels unclear if it would hold for encoder only or multimodal generative systems. This type of restricted generality is one of the weaker parts of the work.

While a variety of models are tested, most comparisons are only between two model scales eg 8b vs 671b which would limit conclusions about scaling trends or arch dependence.

One last minor note is with presentation, the paper is occasionally heavy on technical detail without enough intuitive interpretation of figures or implications.

**Questions:**

Could the authors further elaborate on why II was chosen over more standard representational similarity measures like CKA, cvcca, etc.? II seems to provide an asymmetric view, which is helpful, but would be helpful to know if the location of the ‘semantic layer’ would be the same under a symmetric similarity measure. ie do the layers with minimal II also correspond to peaks in mutual similarity index? Clarifying this would also help cement or strengthen that these layers are truly special, and not simply an artifact of the particular metric. This would also provide additional interpretability of the absolute II values.
A followup is, how precisely is the semantic region defined? Is it by a threshold or by visual inspection of II minima? Could there be the possibility of multiple disjoint semantic pockets?
It is not clear whether these effects generalize to encoder-only or non-autoregressive models like BERT, XLM. Could the authors verify?
Is it possible the discovered semantic alignment is exploited, like by stitching models or improving multimodal transfer?

---

> ### Author Response · Authors · 2025-11-19
>
> We thank the reviewer for their rich feedback.
>
> Reply to weaknesses:
>
> **Methodological advancements:**
>
>
> We focus on how the representation alignment depends on network depth (Fig. 1), on the number of tokens used to represent a sentence (Fig. 2), and on the choice of taking their average or their concatenation (Supp. Inf. E, Fig. 6). In order to handle representational spaces with hundreds of thousands of neurons, which are in general prohibitively large, we took advantage of the fact that the II does not change if one binarizes the representations (Supp. Inf. H, Fig. 9). None of these results was known from any previous contribution that we are aware of, and, in our opinion, they are all important methodological advancements.
>
>
> **Model sizes:**
>
> Further comparison with other sizes is indeed interesting, and was not added because of the lack of space, and our focus on the largest available open-source model, DeepSeek-V3-671B. We have also observed that Qwen7b has a very similar behavior to the models present in our manuscript, and we can add it as a further comparison.
> In the revised version of the manuscript we will emphasize that our results hold for several different language and vision models with heterogeneous architectures, on different datasets, and this, in itself, shows that our similarity measurements are meaningful and robust. We will remove the emphasis on model size, which, as correctly pointed out by the referee, is not systematically studied on its own.
>
> **Causal asymmetry and token correlations:**
>
> We focused on causal models because they are the state of the art for natural language processing. Nonetheless, for comparison, we will add our results on BERT, in which we observed different directional asymmetries. More precisely, BERT representations have a causal asymmetry both in the first and last layers, and they present no asymmetry in deep layers. These results thus suggest that the asymmetries are architecture-dependent, and we are going to add them to our manuscript for completeness.
>
> **Presentation of our results and their technicality:**
>  Could you please be more specific on where this happens, such that we can modify the text accordingly to improve the presentation?
>
>
> **Questions:**
> Could the authors further elaborate on why II was chosen over more standard representational similarity measures like CKA, cvcca, etc.? II seems to provide an asymmetric view, which is helpful, but would be helpful to know if the location of the ‘semantic layer’ would be the same under a symmetric similarity measure. ie do the layers with minimal II also correspond to peaks in mutual similarity index? Clarifying this would also help cement or strengthen that these layers are truly special, and not simply an artifact of the particular metric. This would also provide additional interpretability of the absolute II values. A followup is, how precisely is the semantic region defined? Is it by a threshold or by visual inspection of II minima? Could there be the possibility of multiple disjoint semantic pockets? It is not clear whether these effects generalize to encoder-only or non-autoregressive models like BERT, XLM. Could the authors verify? Is it possible the discovered semantic alignment is exploited, like by stitching models or improving multimodal transfer?
>
>
> Reply to questions:
>
> **Other similarity metrics:**
> In Supplementary Information G we compare our results to the neighborhood overlap from Ref. 3, that was extensively compared to CKA, and was found to be preferable over CKA, which is only a linear method. Both the neighborhood overlap and the II  fully agree on where the representation similarity is maximal, as shown in Fig. 8 from Supp. Information. We will clarify the importance of the results in Supplementary Information G in the revision.
>
> **Interpretation of the absolute values of the II:**
> Section C of Supplementary Information precisely provides such heuristics, connecting it with a percentage of shared features between representations. Such interpretation should help any reader to understand what the value of the II means in this context, and we will clarify this in the revision.
>
> **Semantic layers:**
> The semantic layers are not defined by a threshold but only qualitatively, as the region in the NN in which the II is small. To avoid confusion we will avoid mentioning “semantic layers” in the absence of a well defined threshold to identify them univocally.
>   We will add as perspectives for future work, the comparison of the II values against cross-modality retrieval scores, where we expect the layers with lowest II to have the best retrieval score. Also, as suggested by the referee, we expect those layers to be the best layers to do model stitching.

---

> > ### Author Response · Authors · 2025-11-26
> >
> > **Summary of Revisions**
> >
> > We thank all reviewers for their thoughtful feedback. In the revised manuscript, we have:
> >
> > • **Softened and clarified several claims**, explicitly stating limitations and avoiding over-general interpretations.
> >
> > • **Expanded comparisons across model sizes and architectures**, adding results for LLaMA models of different sizes (1B, 3B, 8B) and for BERT. These additional experiments **validate and confirm our main findings**, showing that the key alignment patterns and asymmetries persist across scales and architectures.
> >
> > • **Clarified key methodological points**, including the relation between Information Imbalance and neighbourhood-based measures, the effect of averaging vs. concatenating tokens, and the consistency between binarised and floating-point representations.
> >
> > • **Improved presentation quality**, fixing citation style, figure notation, and overall clarity. All revisions are highlighted in blue in the updated manuscript.
> >
> > We believe these modifications directly address the reviewers’ critiques and substantially strengthen the work.

---

### Official Review · Reviewer_bJsH · 2025-11-01

**Soundness:** 3
**Presentation:** 2
**Contribution:** 2
**Rating:** 4
**Confidence:** 3

**Summary:**

The paper applies the Information Imbalance metric to study where semantics emerge in large text and vision models. While technically sound and supported claims, the work offers limited novelty. The contribution is clear, but could benefit from stronger contextualization and benchmarking against other methods.

**Strengths:**

1 - Using information imbalance to capture asymmetric predictivity between representation spaces is a good fit for the paper’s goals and is well justified. The paper includes sanity checks and reference experiments to interpret II values.

2 - A solid and reproducible experimental setup is employed, with the II metric applied systematically to multiple large models in various configurations.

3 - The identification of token-distributed semantic encodings, long-range correlations in interior layers, and cross-modal asymmetries is interesting and relevant.

**Weaknesses:**

1 - The paper lacks connection to prior work; while it positions its goal against existing work, it doesn’t compare to any previously introduced methods, correlation measures, probing tasks or feature insights. The omission weakens the claim that this is a novel contribution.

2 - The main observations (semantic information concentrated in middle layers, language-agnostic representations emerging with scale) echo prior findings. The paper does not convincingly show what new understanding II provides beyond known cross-lingual or multimodal analyses.

3 - The authors equate low II with “semantic information” without validating it against a downstream measure of semantics (e.g., translation accuracy, retrieval alignment, or semantic probing). This conceptual step needs empirical justification.

4 - The insights, while valuable, do not sufficiently support the claim “converge towards shared representations of the world”, as correlations and causation are different.

5 - The figures show inconsistencies; no error bars are visible, despite being described in the legend.

**Questions:**

1 - Can you show that II correlates with established semantic metrics such as cross-lingual retrieval accuracy or probe performance?

2 - Why not compare directly with previous work on semantic measures across languages to base the interpretations?

3 - Can you clarify in what sense II reveals new structures that existing analyses could not?

4 - Have you checked whether tokenisation differences across languages/models materially affect the II minima?

---

> ### Author Response · Authors · 2025-11-19
>
> We thank the reviewer for constructive feedback.
>
> **Connection to prior methods and findings.**
>
> The comparison required by the reviewer is already described in Appendix G, and we will point at it more clearly in the revision. In that experiment, we compare our results to the fraction of shared k-nearest neighbours, following the approach of Huh et al. (2024), who evaluate a wide range of representational similarity metrics and find that local neighbour-based measures perform best. Our use of information imbalance (II) extends this line of work by introducing an asymmetric, rank-based measure that directly captures directional discrepancies between representations—an aspect not addressed in Huh et al. We further explore different summary statistics for layers (last token, token chains, mean-pooled tokens) and evaluate all layers rather than selecting the best alignment post hoc.
>
> Our contribution is to connect two observations:
> 1. models develop semantic structure detectable by multiple measures, and
> 1. representations of semantically equivalent inputs tend to converge across models and modalities.
> We show that a single local, asymmetric similarity measure (II) can quantify and localise this convergence across distinct scenarios: (i) translations; (ii) image–caption pairs; and (iii) visually similar objects. Rather than claiming causality, our goal is to provide evidence consistent with a shared-representation hypothesis across diverse conditions.
>
> **Downstream semantics.**
>
> We agree that linking II to semantic behaviour is important. We would like to clarify that several of our experimental settings already constitute semantic evaluations, even if not explicitly described as such:
> - **Translation pairs** require the model to encode meaning across languages; semantically equivalent sentences often have no lexical overlap, so representational convergence necessarily reflects semantic alignment.
> - **Image–caption pairs** form an implicit cross-modal grounding task based on shared semantic information, widely used as a benchmark (e.g., CLIP retrieval).
> - **Image-category** experiments evaluate whether visual representations cluster according to conceptual categories, which can be seen as a semantic criterion.
>
> In all cases, the convergence we measure is driven by semantic equivalence across languages or modalities, not by syntactic or low-level statistical similarity. Thus, these settings provide information about semantic content, as a  downstream semantic task would do. However, we agree with the reviewer that adding a standard semantic downstream benchmark would strengthen the evaluation and we will add a the cross-lingual retrieval experiment—testing whether the layers with lower II also achieve higher top-k retrieval accuracy.
>
> **Formatting.**
>
> We agree that the presentation should be clear and consistent. Regarding error bars: they are included in the figures, but are difficult to see due to their small magnitude. This is explicitly stated in the captions. We will also ensure consistent notation and bibliographic formatting, following the conventions from Goodfellow et al.’s Deep Learning as implemented in the ICLR style files.

---

> > ### Comment · Reviewer_bJsH · 2025-11-25
> >
> > Thank you for your detailed response and clarifications. I appreciate the effort to address the points raised.
> >
> > While I agree that the paper provides evidence consistent with a shared-representation hypothesis, I would still insist that “converge towards shared representations of the world” remains over-claimed. Especially given the results of Figure 4.
> >
> > In terms of prior work, could you position your results alongside other works that quantitatively compare representations across subspaces, such as *Lan et al.* [1] and *Chughtai et al.* [2] ?
> >
> > I still have concerns regarding Q3, Q4 and W2. Could you please elaborate on these?
> >
> > [1] Lan, Michael, et al. "Quantifying Feature Space Universality Across Large Language Models via Sparse Autoencoders." arXiv preprint arXiv:2410.06981 (2024).
> >
> > [2] Bilal Chughtai, Lawrence Chan, and Neel Nanda. 2023. A toy model of universality: reverse engineering how networks learn group operations. In Proceedings of the 40th International Conference on Machine Learning (ICML'23), Vol. 202. JMLR.org, Article 248, 6243–6267.

---

> > > ### Author Response · Authors · 2025-11-26
> > >
> > > Thank you for the questions.
> > >
> > > First, we carefully reviewed the manuscript. We just uploaded the revised version, with the changes highlighted in blue. We softened all statements that could be interpreted as overclaiming, and we added further evidence where appropriate.
> > >
> > > When we refer to a **“shared interpretation of the world,”** we mean the following: if two inputs encode the same underlying concept—for example, two human translations of a sentence, or an image and its caption—then mapping them into a model’s latent space should preserve their conceptual similarity. In an idealised setting, the composition of **“idea → language → latent representation”** should approximate the identity transformation on this conceptual manifold.
> > >
> > > ## How our work differs from [1]
> > >
> > > The goal of [1] is to compare how different LLMs process the same input, using feature directions obtained via sparse autoencoders.
> > > In contrast, **we investigate whether representations of the same concept—translations, captions, images—are encoded in a consistent manner.** In other words, rather than comparing representations of the same string, we compare representations of semantic synonyms in a broad sense.
> > >
> > > We explicitly validate this assumption through a null experiment: **shuffling the pairs destroys all alignment**, confirming that the effect we measure arises from semantic equivalence rather than low-level statistical artefacts.
> > >
> > > We differ from [1] in several further ways:
> > >
> > > - [1] decomposes activations into linear directions and compares models within the same family on identical inputs.
> > >   **In contrast, we study cross-model, cross-modality, and cross-lingual equivalence classes of inputs.**
> > >
> > > - [1] does not examine whether models learn similar features for different encodings of the same concept.
> > >
> > > - [1] employs similarity metrics that [3] have shown to correlate poorly with representational alignment, without discussing why they might still be valid in their setting.
> > >   **Our contribution instead is to provide (i) an asymmetric metric capable of detecting directional effects and (ii) a measure that remains robust at scale.** We cast shared representation as a question of relative information content and ask a simple but fundamental question: **are synonyms represented similarly?**
> > >
> > > ## Relation to [2]
> > >
> > > The work of [2], which we now cite in the revised version, studies universality in a highly controlled toy setting, with networks trained on finite-group composition. While theoretically insightful, its scope is different from ours:
> > >
> > > - we analyse **real large-scale LLMs and ViTs**, and
> > > - datasets that reflect **naturalistic semantics** (translations, captions, images).
> > >
> > > Thus, while conceptually related, their setting abstracts away most of the phenomena—multilinguality, modality gaps, scaling, token structure—that we investigate.
> > >
> > > ## Regarding Q3 (novelty of the metric)
> > >
> > > The novelty of the **information imbalance** lies in its **asymmetry** and the fact that it **does not rely on linearity assumptions** that rarely hold in extremely high-dimensional activations.
> > >
> > > Asymmetry is crucial because we compare:
> > >
> > > 1. different models,
> > > 2. different encodings, and
> > > 3. different tasks.
> > >
> > > This allows us to track:
> > >
> > > - how equivalent inputs evolve across layers, and
> > > - how information is distributed across tokens and layers.
> > >
> > > Importantly, we also show that information imbalance has comparable discriminative power to the neighbourhood-overlap metric—the measure recommended by [3]—while adding **directional information** that existing symmetric measures cannot capture.
> > >
> > > ## Regarding Q4 (tokenisation effects)
> > >
> > > We believe tokenisation does not materially affect our conclusions. We intentionally analyse models with different tokenisers, and the qualitative shape and location of semantic regions remain stable across tokenisation schemes.
> > >
> > > ## Regarding W2 (novelty of workflow and experimental design)
> > >
> > > Our novelty stems from:
> > >
> > > 1. **combining a broader notion of “same input”** with
> > > 2. **an asymmetric metric that directly measures semantic predictability.**
> > >
> > > We systematically test equivalence classes defined by meaning—translations, image–caption pairs, images from the same category—across multiple architectures and model sizes.
> > >
> > > If semantic content drives representation, then these equivalence classes should align regardless of encoding. We then identify where this alignment is maximised and show that it is consistent with the existence of **semantic layers**.
> > >
> > > Finally, we test different summary statistics of the layer representations and demonstrate that the **last token alone does not capture the full semantic signal**, despite the autoregressive objective of LLMs—another observation that, to our knowledge, does not appear in prior work.
> > >
> > > ---
> > >
> > > **Reference [3]**
> > > Minyoung Huh, Brian Cheung, Tongzhou Wang, and Phillip Isola. *Position: The platonic representation hypothesis.* URL: https://proceedings.mlr.press/v235/huh24a.html

---

> > > > ### Comment · Reviewer_bJsH · 2025-11-27
> > > >
> > > > Thank you for taking the time to clarify your position.
> > > >
> > > > I have further concerns regarding the results:
> > > > * In Figure 1, do you have an explanation for the II bump in the last layer of DeepSeek-V3?
> > > > * I think the results of Figure 3a are slightly misinterpreted and should be compared to the results obtained in [1], which studied the English bias of the latent spaces of certain LLMs.
> > > > * The asymmetry in Figure 3b seems anecdotal and does not convey actionable information.
> > > > * Further baselines, like Embedding models trained for cross-lingual retrieval, would strengthen the contribution.
> > > >
> > > > In general, I think the work is interesting and timely, but the results section is dense and challenging to navigate. The introduction of the metric could also be expanded to improve readability and approachability.
> > > >
> > > > [1] Wendler, C., Veselovsky, V., Monea, G., & West, R. (2024). Do Llamas Work in English? On the Latent Language of Multilingual Transformers.

---

> > > > > ### Author Response · Authors · 2025-12-03
> > > > >
> > > > > > In Figure 1, do you have an explanation for the II bump in the last layer of DeepSeek-V3?
> > > > >
> > > > > Given that the last layer has to predict a next token in a specific language, that domain-specific information should not be transferable across languages, raising the II values. We can further comment that the behavior of DeepSeek-V3 is smooth although pronounced, and it looks like a discontinuity only because we are not showing all 61 layers  but a subsample of them. A similar behavior is observed in Llama-3, although much weaker. Importantly, even if the II of the last layer is higher than in middle layers, it is well below the II of the initial layers, suggesting that the prediction of the next token is done by a representation carrying the semantic content generated in previous layers.
> > > > >
> > > > > Concerning this effect, we had briefly commented in Page 4 of our manuscript that
> > > > >
> > > > > “For the very last layers of DeepSeek-V3, we observe a sudden rise in Information Imbalance (II), since those layers are generating the output in a specific language, which cannot be universal.”,
> > > > >
> > > > > but we will add the previous comments for further clarity.
> > > > >
> > > > > > I think the results of Figure 3a are slightly misinterpreted and should be compared to the results obtained in [1], which studied the English bias of the latent spaces of certain LLMs.
> > > > >
> > > > > Qualitatively, we observed that the more correlated the representations in each language are, the lower the Information Imbalance between them is. However, we moved the figure in the Supp. Inf. to improve readability of the manuscript, leaving this qualitative observation out of the main discussion.
> > > > >
> > > > > The paper [1] is indeed very interesting, although they focus solely on logic-lens experiments, and not in representation similarity, nor in the quantification of token-token correlations. Moreover, in Fig. 4 of our present manuscript we observe representation similarities between English sentences processed by a language-only model (DeepSeek-V3) and images processed by vision-only models (Dino and iGPT), suggesting that the similarities we find are not induced by English, but by abstract semantics.
> > > > >
> > > > > We believe that our results are complementary to those of referee's Ref. [1], – cited in “related work” – and we added the previous comments to the final version of our manuscript.
> > > > > The asymmetry in Figure 3b seems anecdotal and does not convey actionable information.
> > > > > We believe the observed asymmetries in correlations were not known in the literature, and they are moreover **impossible** to capture by conventional metrics like CKA, linear correlation between tokens, or the neighborhood overlap from the Platonic Representation paper. We believe it could be very interesting to see if the observed asymmetries can be quantitatively related to downstream task performances. Although, as correctly pointed out by the referee, since these results are not fundamental for the rest of our main manuscript, we moved them to Supplementary Information, Section I.
> > > > >
> > > > > Concerning asymmetries, in the new Figure 1(left), we report a synthetic case that highlights how the Information Imbalance captures asymmetric predictivities between datasets to stress this relevant property of the metric we employ.
> > > > >
> > > > > > Further baselines, like Embedding models trained for cross-lingual retrieval, would strengthen the contribution.
> > > > >
> > > > > We thank the referee for the suggestion. We completely agree that such experiments are interesting and we mention them in Limitations as a due follow-up study.
> > > > >
> > > > > > In general, I think the work is interesting and timely, but the results section is dense and challenging to navigate. The introduction of the metric could also be expanded to improve readability and approachability.
> > > > >
> > > > > We thank the referee very much for the constructive and useful feedback. We believe that the points raised enhanced the clarity of our paper, and how it is positioned among other works. We rewritten the introduction of the metric to enhance clarity and motivation. We also re-organized the presentation of the material, moving from the Supp. Inf. to the main text the current Figure 1(right), adding also CKA for reference.
> > > > >
> > > > >
> > > > > [1] Wendler, C., Veselovsky, V., Monea, G., & West, R. (2024). Do Llamas Work in English? On the Latent Language of Multilingual Transformers.

---

### Official Review · Reviewer_vrsg · 2025-11-01

**Soundness:** 2
**Presentation:** 2
**Contribution:** 2
**Rating:** 2
**Confidence:** 3

**Summary:**

This paper presents a quantitative analysis of semantic information in deep representations of text and images using the Information Imbalance metric. The authors examine how Large Language Models (LLMs) and vision transformers encode semantic content across different modalities, languages, and architectural layers. Key findings include the identification of "semantic layers" where language-transferable information is concentrated, the observation that semantic information is distributed across multiple tokens with long-range correlations, and the discovery of significant model-dependent asymmetries in information content between image and text representations.

**Overall assessment**

This paper makes valuable contributions to understanding semantic representations in LLMs by applying the Information Imbalance metric. However, the work would benefit significantly from addressing the presentation issues and expanding the experimental scope to support the broader claims.

**Strengths:**

* The paper identifies inner "semantic" layers containing the most language-transferable information, significantly extending existing research in the field. The authors examine these inner semantics across tokens and input modalities, revealing significant model-dependent asymmetries between representations.
* The paper presents interesting and suggestive findings across two models as well as across multiple language pairs. These findings are promising and lay the groundwork for more systematic exploration across languages, model sizes, and model families.
* The use of the Information Imbalance metric provides an asymmetric measure of relative information content, which is well-suited for comparing representations across different architectures and modalities.

**Weaknesses:**

* The article lacks polish in several areas, which impacts readability. Issues include misformatted citations, mentions of error bars that are not visualized, and inconsistent use of abbreviations across figures.

* Multiple strong claims are not supported by experiments on only two models of different sizes and architectures. For example, the section titled "CORRELATIONS BETWEEN TOKENS AS A HALLMARK OF QUALITY IN SEMANTIC REPRESENTATIONS" (Anonymous, p. 6) and the claim "We further ascertained that, on these 'semantic' layers, long token spans meaningfully contribute to the representation, and long-distance correlations in encoded information cue high-quality representations." (Anonymous, p. 9). These would benefit from more systematic exploration across model sizes within the same family.
* The exploration across languages relies primarily on a limited set of language pairs. The heterogeneity claims (Section 3.1.1) would be strengthened by more systematic sampling across languages with varying levels of representation in training data.
* The paper would benefit from a more systematic exploration of model size as an independent variable. LLama3.1-8B and DeepSeek-V3 vary on multiple parameters beyond size, making it difficult to isolate the effect of scale on semantic representation quality.

**Questions:**

**Suggestions**

* Please format all citations correctly according to the conference style guide.

* Consider writing out "Information Imbalance" in full rather than using the abbreviation "II," especially given its inconsistent use across figures.

* Figure 2: Error bars are mentioned in the caption but are not visible in the figure. Please either make them visible or revise the caption.

* I would write "opus_books" as "opus books" or "OPUS Books" rather than with an underscore. You already use the citation and appendix D for an exact reference

* The statement "proving that semantic information is not concentrated in the last tokens, but spread over many of them" (Anonymous, p. 5) makes too strong a claim. Consider revising to "demonstrating that" or "showing that" instead of "proving that."

* The conjecture "We conjecture that this difference is due to the greater online presence of the Spanish language relative to Italian Lan (2025), and the consequent difference in the amount of training data." (Anonymous, p. 5) could be examined more systematically across languages based on their presence in Common Crawl or similar corpora. This would strengthen the paper's empirical foundation.

* Consider adding experiments that:
  - Compare multiple models of different sizes within the same family (e.g., Llama3.1 at 8B, 70B, and 405B)
  - Sample more systematically across languages with known differences in training data representation
  - Test whether the identified patterns hold across additional model families

---

> ### Author Response · Authors · 2025-11-19
>
> **We thank the reviewer for the thoughtful comments.**
>
> **Formatting.**
> Regarding the reviewer’s concern about error bars: the the error bars are smaller than the marker size. This is explicitly stated in the figure caption. In the revised version, we  will ensure consistent notation and bibliographic formatting, following the conventions from Goodfellow et al.’s Deep Learning book as included in the ICLR style.
>
> **Model choice.**
> Our goal is to investigate how information is encoded and internally organised across different architectures and modalities. To this end, we study different datasets of (i) book translations; (ii) image–caption pairs; (iii) images from common categories across: (i) two large language models of different sizes; (ii) a self-supervised vision transformer; and (iii) an autoregressive pixel transformer, in three settings—translations, image–caption alignment, and image categories. Similarity is assessed via local neighbourhood relationships between representations. We agree that controlling for a single model family across multiple scales is valuable; however, our study spans several orthogonal degrees of freedom—model size, training objective, and data domain. Given these interacting factors, we intentionally include one representative experiment per scenario to test whether the observed phenomena persist across qualitatively different architectures and tasks.
>
> **Strength of the conclusions.**
> We agree that the conclusions should be phrased with appropriate caution. Our aim is not to claim that all intermediate layers universally encode an abstract notion of semantics. Rather, we show that across distinct models and modalities, representations converge in the layers where prior work suggests semantic information tends to emerge. While we do expect that a vertical focus on model size should highlight increasing degrees of semantical awareness -as per the existing literature- our focus is horizontal. Our main focus is consistency: we investigate the relation between meanings and representations across languages, modalities, and architectures. We agree that both directions are interesting, but given the page constraints of the initial submission, we prioritised breadth across settings over depth within a single model family.

---

> > ### Comment · Reviewer_vrsg · 2025-11-19
> >
> > Thanks for the response.
> >
> > I want to reiterative that I don't argue that the experiments are not extensive, but I argue that they are not sufficient for the claims made. i see reformulating the claims and clearly stating the limitations as a viable strategies.
> >
> > Looking forward to seeing the revised version.

---

> > > ### Author Response · Authors · 2025-11-26
> > >
> > > **Summary of Revisions**
> > >
> > > We thank all reviewers for their thoughtful feedback. In the revised manuscript, we have:
> > >
> > > • **Softened and clarified several claims**, explicitly stating limitations and avoiding over-general interpretations.
> > >
> > > • **Expanded comparisons across model sizes and architectures**, adding results for LLaMA models of different sizes (1B, 3B, 8B) and for BERT. These additional experiments **validate and confirm our main findings**, showing that the key alignment patterns and asymmetries persist across scales and architectures.
> > >
> > > • **Clarified key methodological points**, including the relation between Information Imbalance and neighbourhood-based measures, the effect of averaging vs. concatenating tokens, and the consistency between binarised and floating-point representations.
> > >
> > > • **Improved presentation quality**, fixing citation style, figure notation, and overall clarity. All revisions are highlighted in blue in the updated manuscript.
> > >
> > > We believe these modifications directly address the reviewers’ critiques and substantially strengthen the work.

---

> ### Comment · Reviewer_vrsg · 2025-11-28
>
> After having reviewed the revised version and seeing many of my concerns addressed, I have revised my score.
>
> While many of the desired elements have been added, I still believe that many of the claims rely on experiments across sizes, which are currently in the appendix. I would love to see a future version of this paper incorporate this into the main body.

---

### Author Response · Authors · 2025-12-03
**Final comment to Area Chair**

Dear Area Chair,

 All referees found our results timely and relevant, but raised important concerns on the clarity of the presentation and on the relationship with previous work. Following the discussion with them, which was unfortunately interrupted by the security leak, we modified the manuscript.  We uploaded a final version that, we believe,  addresses all their concerns. In particular:

> We added to Figure 2 (previously Figure 1), three extra models: two extra sizes of Llama, and a BERT model. All show qualitatively the same behavior, but, as suggested by the reviewers, we find a significant dependence on the  model-size  with fixed architecture. We also find that the results are robust with respect to the change of  the  training objective.

> We moved to Supplementary Information I the previous Figure 3, to keep the discussion of the main article focused on quantifying semantic alignment between representations, and the importance of considering many tokens.

>We rewritten Section 2, where we introduce the metric we use. We added the current Figure 1, which provides two simple synthetic data examples to illustrate  the advantages of using our metric, the Information Imbalance, rather than CKA, a standard alignment metric. We stress that CKA  (i) cannot capture asymmetric information content between representations, and (ii) that it performs poorly in high dimensions.

We thank all the reviewers for the useful feedback, that greatly improved the presentation of our manuscript.

---

### Meta-Review · Area_Chair_U97o · 2026-01-10

**Summary:**

This paper introduces Information Imbalance (II), an asymmetric metric for analyzing how semantic information is encoded in deep representations of vision and language models. Reviewers agree that the topic is timely and interesting, while experimental evaluation is technically sound. However, substantial concerns remain regarding the novelty of the contribution relative to prior work, the strength of claims about “semantic representations” and “shared internal spaces”, the lack of systematic validation across model scale and downstream semantic tasks. Although the author response improves presentation, clarifies the metric and adds experiments on additional models, these revisions do not resolve the limitations fully and the improved presentation would require another round of reviews. So, the paper may not be accepted for ICLR. It is suggested that the authors fully incorporate the review suggestions towards a future submission.

**Reviewer Concerns:**

### Addressed concerns
* **vrsg, qzxA:** Presentation and clarity of the Information Imbalance metric. This is partially addressed by the author response which rewrites the metric description, adds examples, fixes figures and citations.

### Unaddressed concerns

* **bJsH, qzxA:** Limited novelty relative to prior work. The author response emphasizes the asymmetry and properties of II, but does not demonstrate qualitatively new insights beyond existing alignment analyses and the extent of methodological contributions or their treatment in the manuscript remains small.

* **bJsH, GtuN:** Lack of semantic validation. The author response argues that translations, image-caption pairs, and category groupings are inherently semantic and proposes future experiments, but does not add direct validation against downstream semantic tasks (like retrieval, probing).

* **bJsH, GtuN:** Over-interpretation of asymmetries and claims of “shared representations of the world.” Although language was softened, the empirical support remains correlational and not strongly convincing.

* **vrsg:** Claims about general semantic structure are insufficiently supported. This is partially addressed as additional model experiments are added, but validation remains limited and a more extensive rewrite might be needed to match claims to results.

**Reviewer Scores:**

* **vrsg:** Initial rating 2, some concerns are addressed but some remain, would likely raise to 4.

* **bJsH:** Initial rating 4, key concerns remain, would likely maintain 4.

* **qzxA:** Initial rating 4, some presentation issues addressed but methodological ones remain, would likely maintain 4.

* **GtuN:** Initial rating 4, key concerns remain unaddressed, would likely maintain 4.

---

### Decision · Program_Chairs · 2026-01-26

Reject